# LANO: Large Language Models as Active Annotation Agents for Open-World Node Classification

## Abstract

Node classification is a fundamental task in graph learning. While Graph Neural Networks (GNNs) have achieved remarkable success in this area, their effectiveness relies heavily on large amounts of high-quality labels, which are costly to obtain. Moreover, GNNs are typically developed under a closed-world assumption, where all nodes belong to a fixed set of categories. In contrast, real-world graphs follow an open-world setting, where newly emerging nodes often stem from out-of-distribution (OOD) classes, making it challenging for GNNs to generalize. Motivated by the strong zero-shot reasoning and generalization ability of Large Language Models (LLMs), we propose **LANO** (LLMs as Active Annotation Agents for Open-World Node Classification). Our framework first aligns GNN representations with LLM token embeddings via instance-aware and feature-aware self-supervised learning, enabling LLMs to serve as zero-shot predictors for graph tasks. **LANO** then employs an influence- and uncertainty-driven strategy to select the most representative nodes and leverages LLMs for cost-effective pseudo-label generation. To suppress the spread of inaccurate labels and mitigate labeling bias, a soft feedback propagation mechanism disseminates bias-reduced pseudo labels to neighboring nodes with label decay mechanism, followed by iterative GNN optimization. Extensive experiments on multiple benchmarks demonstrate that **LANO** consistently outperforms popular baselines, showcasing the great potential of LLMs as active annotation agents for advancing open-world graph learning.

## 1 Introduction

Node classification is one of the most typical research directions in graph analysis (Xiao et al., 2022), with broad applications in citation network, amazon networks, and recommender systems. Under the closed-world assumption, graph neural networks (GNNs) have achieved remarkable success in this task (Wang et al., 2024b). Despite their effectiveness, GNN-based models face several inherent limitations. *First*, they are notoriously label-hungry—their performance heavily relies on abundant high-quality labeled data, as shown in Figure 1, which is often costly and labor-intensive to obtain (Chen et al., 2023). *Second*, most existing models assume that labeled and unlabeled nodes come from the same set of predefined categories. However, this assumption rarely holds in real-world open-world scenarios, where newly added nodes may belong to entirely novel, out-of-distribution (OOD) categories. As a result, models trained solely on seen classes struggle to generalize to unseen categories, severely restricting their applicability in open-world graph learning (Wang et al., 2024b).

To address this challenge, prior work has explored OOD detection and open-world learning on graphs. Energy-based approaches replace softmax confidence with energy functions to distinguish in-distribution (ID) from OOD nodes (Liu et al., 2020). Other efforts, such as ORCA (Cao et al., 2021), design joint objectives for classification and clustering to progressively discover novel categories, while OODGAT (Song & Wang, 2022) explicitly models interactions between ID and OOD nodes via attention. Although these methods can mitigate misclassification and detect unknown nodes, their performance often degrades significantly under distributional shifts, limiting their generalization capacity in truly open-world environments (Li et al., 2022).

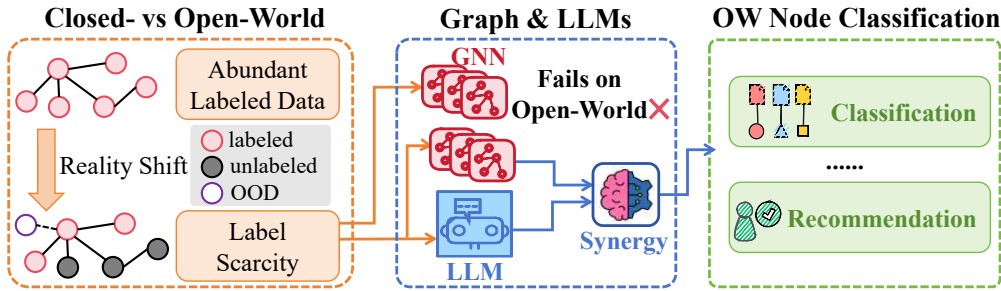

Figure 1: GNN–LLM Synergy for Open-World (OW) Node Classification. Traditional GNNs rely on abundant labeled data and closed-world assumptions, while open-world graphs face label scarcity and the emergence of OOD nodes. Faced with such challenges, the integration of GNNs and LLMs with strong generalization capabilities demonstrates success in OW node classification.

Recent advances in Large Language Models (LLMs) shed light on addressing the challenge. LLMs exhibit remarkable zero-shot ability, often achieving strong performance without requiring labeled data (Chen et al., 2023). Thus as illustrated in Figure 1, LLMs provide a promising direction to alleviate both label scarcity and OOD generalization issues. Compared with costly human anno­tation, LLM-assisted labeling substantially reduces supervision cost. However, directly applying LLMs to graph-based node classification remains challenging: (1) *Structure Awareness*: LLMs are not inherently designed to capture the relational and structural information present in graphs (Wang et al., 2024a); (2) *Node Selection*: annotating all nodes with LLMs is infeasible, making it essential to identify the most representative nodes for labeling; and (3) *Label Reliability*: LLM-generated pseudo-labels are susceptible to hallucinations and biases, requiring mechanisms that can mitigate noise while amplifying their benefits for GNN training (Sheng et al., 2025).

In this paper, we propose **LANO**, a novel framework that leverages LLMs as active annotation agents for open-world node classification. Specifically, our method first employs instance-aware graph learning to learn embeddings from unlabeled nodes, and introduces feature-aware self-supervised alignment to map GNN representations into the LLM token embedding space, thereby enabling LLMs to serve as zero-shot predictors for graph tasks. To further reduce annotation cost, LANO computes node influence and uncertainty to select the most representative nodes for LLM labeling. The obtained pseudo-labels are then propagated to neighboring nodes via a soft label propagation mechanism with label decay, which not only improves efficiency but also mitigates bias in pseudo-labeling. Finally, GNN training is iteratively refined using the enriched supervision, enhancing performance in both ID and OOD settings.

Our main contributions are summarized as follows: (1) *New Perspective*: We introduce a novel per­spective that leverages LLMs with strong zero-shot abilities as active annotation agents to address label scarcity and OOD challenges in open-world node classification. (2) *New Framework*: We propose **LANO**, which integrates GNN–LLM representation alignment, influence- and uncertainty-based node selection, and bias-reduced soft label propagation into a unified framework to iteratively optimize GNN training. (3) *Experiments*: Extensive experiments across multiple datasets demon­strate that our framework consistently outperforms strong baselines, achieving extraordinary results in open-world node classification.

## 2 PRELIMINARIES

**Notations.** We define the graph as $\mathcal{G} = (\mathcal{V}, \mathcal{E}, \boldsymbol{X}, \boldsymbol{A})$, where $\mathcal{V}$ is the set of nodes, $\mathcal{E}$ is the set of edges, $\boldsymbol{X}$ denotes the initial node features, and $\boldsymbol{A}$ is the adjacency matrix which satisfies $\boldsymbol{A}[i, j] = 1$ if an edge exists between nodes $v_i$ and $v_j$. Let $|\mathcal{V}| = N$ be the total number of nodes. The node set $\mathcal{V}$ is partitioned into a labeled subset $\mathcal{V}_l$ and an unlabeled subset $\mathcal{V}_u$. Let $\mathcal{C}_l$ denote the set of classes corresponding to the labeled nodes $\mathcal{V}_l$ (i.e., the set of known classes), and let $\mathcal{C}_u$ denote the set of classes corresponding to the unlabeled nodes $\mathcal{V}_u$. In an open-world semi-supervised learning

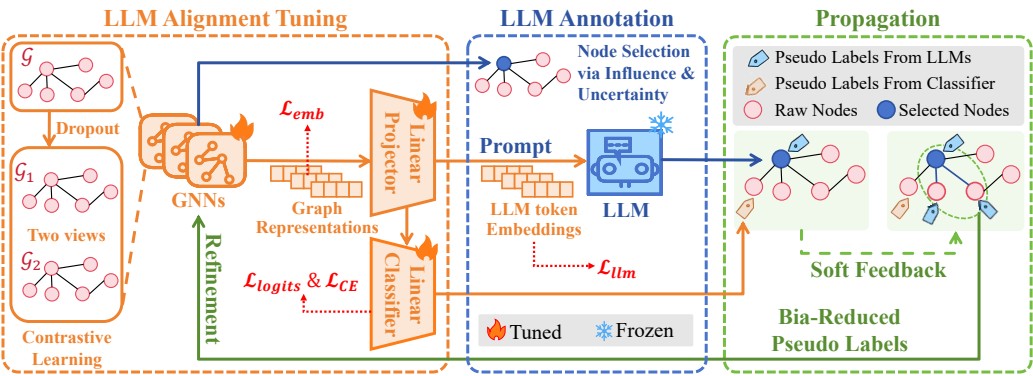

Figure 2: The overall **LANO** framework. After aligning GNN representations with LLM embeddings, LANO selects the most representative nodes for LLM annotation based on influence and uncertainty, thereby generating pseudo-labels. These pseudo-labels are further propagated through soft feedback, and the debiased labels are ultimately leveraged to iteratively optimize the training of the GNN.

setting, $\mathcal{C}_l \neq \mathcal{C}_u$ and they share some overlap, i.e., $\mathcal{C}_l \cap \mathcal{C}_u \neq \emptyset$. The set of novel classes is given by $\mathcal{C}_n = \mathcal{C}_u \setminus \mathcal{C}_l$. The set of manual labels for nodes in $\mathcal{V}_l$ is denoted by $\mathcal{Y}_l = \{y_i \mid v_i \in \mathcal{V}_l, y_i \in \mathcal{C}_l\}$.

**Graph Neural Networks.** Graph Neural Networks (GNNs) are widely used for learning representations of nodes in graph-structured data. The central idea is to iteratively aggregate information from a node's neighborhood to capture both structural and feature dependencies. Formally, a generic GNN layer is given by:

$$\boldsymbol{h}_{v_i}^{l+1} = update\big(\boldsymbol{h}_{v_i}^l,\ aggregate(\{\boldsymbol{h}_{v_j}^l \mid v_j \in \mathcal{N}(v_i)\})\big), \tag{1}$$

where $\boldsymbol{h}_{v_i}^l$ is the representation of node $v_i$ at the $l$-th layer, and $\mathcal{N}(v_i)$ is its neighborhood. The $aggregate$ function summarizes information from neighbors (e.g., mean, sum, or attention-based weighting), while $update$ refines the representation, often via multilayer perceptrons (MLPs) (Sheng et al., 2025). By stacking multiple layers, nodes can capture higher-order structural information. After $L$ layers, the final representation $\boldsymbol{h}_{v_i}^L$ is obtained, which can be applied to downstream tasks.

**Problem Definition.** We investigate the task of node classification in an open-world graph setting (Wang et al., 2024b). Given the graph $\mathcal{G} = (\mathcal{V}, \mathcal{E}, \boldsymbol{X}, \boldsymbol{A}, \mathcal{Y}_l)$, the node set $\mathcal{V}$ consists of a labeled subset $\mathcal{V}_l$ with labels $\mathcal{Y}_l$, and an unlabeled subset $\mathcal{V}_u$. Among them, the unseen labels can be represented as $\mathcal{Y}_u = \{y_i \mid v_i \in \mathcal{V}_u, y_i \in \mathcal{C}_u\}$. The objective is to learn a function

$$\mathcal{F} : \mathcal{G} = (\mathcal{V}, \mathcal{E}, \boldsymbol{X}, \boldsymbol{A}, \mathcal{Y}_l) \longrightarrow \mathcal{Y}_u, \tag{2}$$

such that nodes belonging to $\mathcal{C}_l$ are accurately assigned to their corresponding seen classes, while nodes from $\mathcal{C}_n$ are not only detected as unknown but further distinguished and categorized into their respective novel classes. This formulation extends traditional closed-world node classification by requiring the model to handle both recognition and classification of previously unseen classes within the same unified framework.

## 3 METHOD

We propose a novel semi-supervised graph learning framework, **LANO**, which leverages LLMs as active agents to facilitate open-world node classification. The framework consists of three key modules: (1) *Graph Self-supervised Learning for LLM Alignment*, (2) *Influence and Uncertainty Maximization-aware LLM Annotation*, and (3) *Learning with Bias-Reduced Pseudo Labels*. An overview of the framework is illustrated in Figure 2.

### 3.1 GRAPH SELF-SUPERVISED LEARNING FOR LLM ALIGNMENT

In **LANO**, GNNs encoder is used to generate node representations, while an LLM, owing to its strong generalization ability, serves as a zero-shot predictor for node classification (Wang et al.,

2024a). However, LLMs cannot directly process graph-structured data. To bridge this gap, we propose a self-supervised alignment scheme that maps GNN representations into the LLM token embedding space. Our approach combines *instance-aware* graph learning and *feature-aware* alignment to capture both structural invariance and semantic compatibility.

**Instance-aware Graph Learning.** To reduce reliance on labeled data and facilitate the subsequent classification of OOD, we enhance the model's generalization ability through instance-aware graph learning. Following SimCSE (Gao et al., 2021), we feed the same sample into the feature encoder twice. Due to the stochastic dropout masks applied during each forward pass, the encoder produces slightly different intermediate activations and outputs, while preserving the same underlying structural information. As a result, we obtain two distinct views derived from the same instance—forming a positive pair for the contrastive learning objective. Formally, given the original graph $\mathcal{G}$, we generate two views $\mathcal{G}_1$ and $\mathcal{G}_2$. Each view is encoded by the GNN to produce node embedding matrices:

$$U_* = f_{\text{GNN}}(\mathcal{G}, \ d_*) \in \mathbb{R}^{N \times F_U}, * \in \{1, 2\}, \tag{3}$$

where $d_1$ and $d_2$ denote the dropout masks used in the two forward passes, with $d_1 \neq d_2$, and $F_U$ is the dimension size of node representations. For node $v_i$, we denote its embeddings from the two views as $\boldsymbol{w}_i$ and $\boldsymbol{w}'_i$, respectively. And let $N_a$ denotes the number of randomly sampled nodes in a mini-batch. Representations of the same node under different views are regarded as a positive pair, while those of different nodes are treated as negative pairs. The model learns discriminative embeddings in an unsupervised manner by maximizing the consistency between positive pairs and minimizing the similarity between negative pairs. We adopt a contrastive objective that encourages representations of the same node across views to be close, while pushing apart different nodes. The corresponding loss function is defined as:

$$\mathcal{L}_{emb} = -\frac{1}{2N_a} \sum_{i=1}^{2N_a} \log \frac{\exp(\text{sim}(\boldsymbol{w}_i, \boldsymbol{w}'_i/\tau))}{\sum_{j=1}^{N_a} \mathbf{1}_{[j \neq i]} \exp(\text{sim}(\boldsymbol{w}_i, \boldsymbol{w}_j/\tau)) + \sum_{j=1}^{N_a} \exp(sim(\boldsymbol{w}_i, \boldsymbol{w}'_j/\tau))}, \tag{4}$$

where $sim(\cdot)$ denotes cosine similarity, $\tau$ is the temperature parameter, and $\mathbf{1}_{[j \neq i]}$ is an indicator function that takes the value 1 if $j \neq i$, and 0 otherwise.

**Feature-aware LLM Alignment Tuning.** To bridge the gap between GNN node representations and the semantic space of LLM token embeddings, it is essential to ensure that when GNN outputs are converted into a sequence of token embeddings and provided as prompts, the LLM can perform zero-shot reasoning. This is achieved in two steps: *first*, a feature-aware contrastive alignment is conducted to align the feature axes (columns) of the GNN with the LLM token space; *second*, a linear projector is trained to map the GNN's central node representations into LLM token embeddings. The projector is fine-tuned for alignment while keeping the LLM frozen. Through this feature-aware LLM alignment tuning, GNN outputs are directly adapted to the LLM embedding space, thereby improving generalization across tasks and datasets. The corresponding loss function is calculated as:

$$\mathcal{L}_{llm} = -\frac{1}{2F_U} \sum_{i=1}^{2F_U} \log \frac{\exp(\text{sim}(\boldsymbol{m}_i, \boldsymbol{n}_i)/\tau)}{\sum_{j=1}^{F_U} \mathbf{1}_{[j \neq i]} \exp(\text{sim}(\boldsymbol{m}_i, \boldsymbol{m}_j)/\tau) + \sum_{j=1}^{F_U} \exp(\text{sim}(\boldsymbol{m}_i, \boldsymbol{n}_j)/\tau)]}, \tag{5}$$

where $\boldsymbol{m}_i$ and $\boldsymbol{n}_i$ are the $i$-th feature vectors from two augmented embeddings $U_1$ and $U_2$. Then the overall self-supervised objective is a weighted combination:

$$\mathcal{L}_{\text{Sup}} = \sum_{i=1}^{N_G} (\lambda_1 \mathcal{L}_{emb} + \lambda_2 \mathcal{L}_{llm}) + \mathcal{L}_{logits}, \tag{6}$$

where $\lambda_1, \lambda_2$ balance the two terms, $N_G$ denotes the number of GNNs, and $\mathcal{L}_{logits}$ denotes the supervised contrast loss introduced on the output logits of the classification layer, which is used to directly improve the differentiation of the final prediction results.

Consequently, neither the GNN nor the LLM requires task-specific fine-tuning. Instead, by mapping graph representations into token embeddings via the projector and feeding them into a designed LLM prompt template, the framework enables cross-task and cross-dataset reasoning in a zero-shot manner (Wang et al., 2024a).

## 3.2 Influence and Uncertainty Maximization-aware LLM Annotation

High-quality annotations are crucial for graph learning, yet manual labeling is prohibitively expensive. In scenarios with scarce and noisy labels, it is therefore essential to select the most informative nodes for annotation, striking a balance between performance improvement and labeling cost. To this end, we propose to leverage node influence and uncertainty for guiding LLM-based annotation, and further design task-specific prompts that enable LLMs to effectively capture graph information for zero-shot node classification.

**Influence with Uncertainty Estimation for Node Selection.** Building on reliable influence-based active learning (Zhang et al., 2021), we adopt a joint strategy that combines uncertainty estimation with influence maximization to identify representative nodes for annotation. We first compute the global uncertainty of each node by considering its relation to cluster centroids obtained via $k$-means and neighbor propagation. Specifically, a Student-$t$ distribution is used to assign each node a probability over clusters, and the entropy of this distribution quantifies the uncertainty. Let $\mathcal{Z}$ denote as node representations, this uncertainty is further refined via neighbor aggregation:

$$u(v_i) \;=\; u(v_i) \;+\; \frac{1}{|\mathcal{N}(v_i)|} \sum_{v_j \in \mathcal{N}(v_i)} \text{sim}(\boldsymbol{z}_i, \boldsymbol{z}_j) \cdot u(v_j), \tag{7}$$

where $\mathcal{N}(v_i)$ denotes the $k$-nearest neighbors of $v_i$, $\boldsymbol{z}_i$ is the node representation, and $\text{sim}(\cdot, \cdot)$ is a similarity function. Next, we estimate the influence of each node by considering multi-hop propagation paths (self, 1-hop, and 2-hop neighbors). Following RIM (Zhang et al., 2021), the influence score from node $v_i$ to $v_j$ after $k$ propagation steps is defined as:

$$Q(v_j, v_i, k) \;=\; r_{v_i} \cdot I(v_j, v_i, k), \tag{8}$$

where $r_{v_i}$ denotes the influence quality of node $v_i$, whose calculation is given in Appendix H, and $I(\cdot)$ measures the reliable influence of $v_i$ on $v_j$ after $k$-step label propagation. Combining global uncertainty and influence, the selection score for node $v_i$ is given by:

$$\text{score}(v_i) = u(v_i) \cdot \sum_{v_j \in \mathcal{N}_k(v_i)} Q(v_j, v_i, k), \tag{9}$$

and the top-$K$ nodes are selected for LLM annotation, the set of selected nodes is denoted as:

$$S = \arg \text{topK}_{v \in \mathcal{V}} \, \text{score}(v). \tag{10}$$

**Prompt Engineering for Annotation.** The prompts are structured into three components: task information, graph information, and output rules. The task information is expressed as a question + option set (Wang et al., 2024a); the graph information consists of node graph token embeddings; and the output rules specify the classification result and confidence. For example: *Your task: Classify the target node into predefined categories or detect a new category. Predefined categories (represented by semantic tokens): category 1: ... Target node: ... Rules: Known/New/Uncertain Category and Confidence Level.* The complete prompt design and motivation is provided in Appendix D.

After identifying the nodes requiring annotation by the LLM, we first align them to the LLM's semantic space and construct carefully designed prompts as input. The output from the LLM falls into three categories: (1) If the LLM outputs a seen (known) class label, we assign a soft label as a confidence-weighted one-hot vector; (2) If the LLM outputs an unseen (novel) class, we use the classification head's predicted distribution, also weighted by confidence; (3) If the output is invalid or malformed, the result is discarded. We retain only valid annotations and discard outdated nodes to ensure high-quality supervision in the subsequent training stage.

## 3.3 Learning with Bias-Reduced Pseudo Labels

Although LLMs can provide pseudo-labels of the selected nodes via carefully designed prompts, these annotations are not guaranteed to be correct. To mitigate the risk of propagating noisy pseudo-labels and mitigate labeling bias, we introduce two mechanisms: *soft feedback propagation* and *bias-reduced pseudo label concordance*.

**Soft Feedback Propagation.** To efficiently expand the utility of LLM-generated pseudo-labels, we propagate them to structurally and semantically similar neighbors. However, directly propagating

hard labels risks amplifying errors. To address this, we adopt a soft feedback propagation strategy. Specifically, the LLM first provides initial annotations for a subset of nodes, where only high-confidence outputs from the LLM are allowed to propagate. These labeled nodes then propagate their labels through the graph structure to generate soft label distributions for neighboring nodes. For each node $v_i^s$ in the selected set $S$, its LLM-generated pseudo-label $y_i^{\text{LLM}}$ is propagated to uncertain neighbors $v_j$ based on feature similarity:

$$s_j = (1 - \text{sim}(z_j, z_i^s)) \cdot s_j \; + \; \text{sim}(z_j, z_i^s) \cdot y^{\text{LLM}}, \tag{11}$$

where $s_j$ denotes the soft prediction vector of node $v_j$, $\text{sim}(\cdot, \cdot)$ measures the similarity between embeddings $z_j$ and $z_i^s$, and $y^{\text{LLM}}$ is a one-hot vector.

During this process, if an uncertain neighbor node $v_j$ is assigned to the same cluster as the LLM-labeled node $v_i^s$ based on the $s_j$, the pseudo label $y_i^{\text{LLM}}$ will be propagated to $v_j$. Otherwise, if the propagated soft label distribution of the labeled node deviates significantly from its original LLM label, the annotation is either rejected or down-weighted. Therefore, pseudo-labels are propagated only when the updated prediction of the node is consistent with the LLM-assigned class, reducing the risk of error amplification. The propagated pseudo-label for $v_j$ is then assigned as:

$$y_j^{\text{prop}} = \begin{cases} y_i^{\text{LLM}}, & \text{if } \text{argmax}(s_j) = y_i^{\text{LLM}}, \\ -1, & \text{otherwise.} \end{cases} \tag{12}$$

In this way, the soft feedback propagation strategy effectively prevents the accumulation and diffusion of LLM errors across the graph while maintaining high consistency in confident labels.

**Bias-reduced Pseudo Label Concordance.** Since the model is not pretrained, early-stage pseudo-labels are often noisy. To alleviate bias accumulation, we introduce a label decay mechanism that gradually attenuates the influence of outdated pseudo-labels. At each iteration, a batch of $B$ new pseudo-labels is generated by LLMs, and only the most recent $R$ labels are preserved for propagation together with the original labeled set. The historical pseudo-labels are maintained across iterations but scaled down by a decay factor $\gamma < 1$:

$$\hat{\mathcal{Y}}^t = \gamma \cdot \hat{\mathcal{Y}}^{t-1} + \hat{\mathcal{Y}}_{\text{new}}^t, \tag{13}$$

where $\hat{\mathcal{Y}}^t$ denotes the aggregated pseudo-label matrix at iteration $t$, and $\hat{\mathcal{Y}}_{\text{new}}^t$ denotes newly generated pseudo-label matrix at iteration $t$. This decay ensures that earlier noisy annotations gradually vanish, while recent high-quality LLM feedback dominates the training process. Together, soft feedback propagation and bias-reduced concordance mitigate the risks of error amplification and label bias, enabling the model to effectively exploit LLM annotations in an open-world setting (Liang et al., 2024; Wang et al., 2024b).

### 3.4 OVERALL OPTIMIZATION

To optimize GNN training, we incorporate bias-reducing pseudo labels into iterative GNN training. Our loss function during training primarily includes: (1) Supervised Contrastive Loss ($\mathcal{L}_{\text{Sup}}$), designed to better separate seen and unseen classes. (2) Cross-Entropy Loss ($\mathcal{L}_{\text{CE}}$), used to learn valuable manual labels. We update the model using the following overall loss formula:

$$\mathcal{L}_{\text{LANO}} = \eta \mathcal{L}_{\text{CE}} + \mathcal{L}_{\text{Sup}}. \tag{14}$$

where $\eta$ is the scaling factor. The $\mathcal{L}_{\text{Sup}}$ is applied to GNN-output embeddings, projected LLM embeddings, and classification layer logits. This encourages similar samples (different perspectives from the same node) to cluster closely in embedding space while keeping dissimilar samples apart. The $\mathcal{L}_{\text{CE}}$ calculated from classification head logits is used for labeled training nodes, ensuring the model correctly classifies known classes.

## 4 EXPERIMENTS

### 4.1 EXPERIMENTAL SETUP

**Datasets.** We evaluate our method on several commonly used benchmark datasets for node classification tasks, including Citeseer (Kipf, 2016), Amazon_photos (Shchur et al., 2018), Amazon

Table 1: Performance comparison on different datasets under open-world settings with test accuracy (%). **The best results** in each column are highlighted in bold and pink, the second-best results in each column are highlighted in blue.

| Method | Citeseer | | | Coauthor_CS | | | Coauthor_phy | | | Amazon_photos | | | Amazon_computers | | |
|---|---|---|---|---|---|---|---|---|---|---|---|---|---|---|---|
| | all | seen | novel | all | seen | novel | all | seen | novel | all | seen | novel | all | seen | novel |
| OODGAT | 46.4 | 56.9 | 37.5 | 68.1 | 68.8 | 65.6 | 68.3 | 69.4 | 62.5 | 63.0 | 71.1 | 54.5 | 61.3 | 63.3 | 55.9 |
| OpenWGL | 62.4 | 71.0 | 54.2 | 58.6 | 67.1 | 50.3 | 73.3 | 85.0 | 68.1 | 71.8 | 74.8 | 69.3 | 57.6 | 65.9 | 44.6 |
| ORCA-ZM | 58.3 | 72.8 | 44.4 | 75.0 | 74.2 | 73.5 | 64.7 | 81.1 | 55.9 | 74.6 | 89.9 | 58.2 | 63.8 | 73.7 | 52.6 |
| ORCA | 58.2 | 68.0 | 49.0 | 73.9 | 81.6 | 68.3 | 66.2 | 84.8 | 58.2 | 76.2 | 87.1 | 64.9 | 60.9 | 67.8 | 53.7 |
| SimGCD | 61.5 | 70.6 | 53.4 | 71.2 | 84.2 | 61.2 | 60.9 | 81.1 | 52.8 | 80.5 | 90.0 | 70.8 | 61.9 | 73.8 | 50.3 |
| OpenLDN | 62.3 | 73.9 | 51.6 | 68.4 | 80.6 | 60.3 | 62.2 | 72.4 | 57.2 | 80.9 | 90.6 | 71.9 | 63.3 | 76.5 | 51.8 |
| OpenCon | 68.8 | 75.0 | 62.1 | 73.5 | 83.4 | 67.5 | 65.8 | 95.0 | 55.4 | 82.6 | 92.1 | 72.8 | 62.3 | 74.9 | 51.2 |
| OpenCon | 66.7 | 73.7 | 60.0 | 71.0 | 81.9 | 64.8 | 62.6 | 83.8 | 54.4 | 82.9 | 87.9 | 78.1 | 59.4 | 69.0 | 53.2 |
| InfoNCE | 68.1 | 70.7 | 65.2 | 72.2 | 72.8 | 72.7 | 60.6 | 58.1 | 60.2 | 76.3 | 78.5 | 75.1 | 56.1 | 51.3 | 59.1 |
| InfoNCE+SupCon | 68.1 | 71.9 | 64.1 | 75.6 | 80.3 | 72.0 | 56.3 | 52.5 | 58.9 | 72.4 | 75.1 | 71.0 | 60.5 | 59.7 | 59.8 |
| InfoNCE+SupCon+CE | 68.1 | 73.6 | 62.6 | 76.4 | 80.5 | 72.9 | 55.8 | 54.7 | 56.5 | 74.4 | 77.1 | 73.0 | 62.8 | 79.4 | 56.1 |
| OpenIMA | 68.1 | 71.8 | 64.3 | 77.1 | 78.3 | 75.9 | 78.0 | 93.6 | 72.2 | 83.6 | 89.9 | 77.3 | 67.8 | 77.8 | 59.0 |
| **LANO (Ours)** | **70.2** | 73.8 | **66.2** | **83.4** | **85.2** | **80.7** | **80.2** | 79.6 | **72.6** | **84.3** | 86.2 | **83.1** | **70.3** | 70.4 | **70.2** |

Computers (Shchur et al., 2018), Coauthor_CS (Shchur et al., 2018) and Coauthor_Physics (Shchur et al., 2018). More detailed statistics of the datasets are provided in Appendix B.

**Evaluation Metric.** Under the open-world setting, node categories are divided into seen and unseen classes. A prediction is considered correct only if the model assigns the node to its ground-truth label. For LLM-based annotation, we additionally provide two options—decidable and undecidable. A prediction is counted as correct if the LLM selects decidable and its output matches the true label; if undecidable is chosen, the annotation is regarded as invalid and excluded from accuracy computation. The more details of the metric is given in Appendix E. All experiments are repeated 10 times with different splits, and the reported accuracy is averaged across runs.

**Baselines.** We compare our method against a broad set of baselines applicable to open-world node classification. These include open-world node classification algorithms OODGAT (Song & Wang, 2022) and OpenWGL (Wu et al., 2021), as well as baseline methods for end-to-end open-world semi-supervised learning, namely ORCA (Cao et al., 2021), ORCA-ZM (Cao et al., 2021), SimGCD (Wen et al., 2023), OpenLDN (Rizve et al., 2022), OpenCon (Sun & Li, 2022), InfoNCE (Oord et al., 2018), and OpenIMA (Wang et al., 2024b). More detailed descriptions of these methods can be found in Appendix F.

**Implementation Details.** Our model builds upon the architecture of OpenIMA by extending its original GNN backbone. Specifically, we treat a single GNN network as one head responsible for encoding a single token, and the number of heads is determined by the number of tokens required as input to the LLM. To capture diverse structural representations of the graph, different heads are designed with variations in hop size, hidden dimensionality, number of attention heads, dropout rate, and whether residual connections are applied. To align the input dimension of the projection layers, we set the output dimension of all heads to 256. We employ Adam as the optimizer with a batch size of 4096. Training is conducted over 40 epochs with a learning rate of 0.004. For detailed model and parameter configurations, please refer to Appendix G, and the source code is available at https://anonymous.4open.science/r/C86B.

## 4.2 RESULTS AND ANALYSIS

Table 1 presents the classification accuracy of our method and various baselines on both seen and unseen classes under the open-world setting. Overall, across most datasets and evaluation scenarios, our approach outperforms all competing methods in terms of overall accuracy and unseen-class recognition, while maintaining strong competitiveness on seen-class classification. We attribute these performance gains to several key factors. *First*, incorporating LLMs as pseudo-label generation agents aligns the semantic information encoded by GNNs and leverages the LLMs' semantic discrimination capability, enabling more efficient and accurate annotation of unseen classes and providing more valuable supervision signals for training. *Second*, by combining node influence and uncertainty metrics, we selectively propagate pseudo-labels only to the most representative unlabeled nodes, effectively reducing noise interference and enhancing the reliability and diversity of

pseudo-labels. *Third*, the original framework relies solely on selecting the top $\rho\%$ of nodes closest to cluster centroids as pseudo-labels, which can lead to label oscillations and unstable training when decision boundaries are still ambiguous.

To address this, we introduce soft feedback propagation and label decay mechanisms, mitigating the spread of biased pseudo-labels and reducing the negative impact of oscillations. *Fourth*, our LANO also performs well on larger datasets such as ogbn-arxiv, as illustrated in Table 2, concluding that LANO remains effective and stable across datasets of different scales. We adopt a random-walk–based sampling strategy combined with mini-batch clustering, enabling efficient training and representation alignment for larger graphs. *Finally*, as suggested by theoretical insights from the training framework, the introduction of LLMs partially alleviates the imbalance of supervision signals between seen and unseen classes, further enhancing overall classification performance.

Table 2: Evaluation on a larger dataset ogbn-arxiv by overall test accuracy (%). The best results for each dataset column are highlighted in bold and pink.

| Method | ogbn-arxiv | | |
|---|---|---|---|
| | **all** | **seen** | **novel** |
| ORCA | 41.6 | 44.7 | 34.6 |
| OpenCon | 32.2 | 31.8 | 31.6 |
| OpenIMA | 43.6 | **49.2** | 32.9 |
| **LANO** (Ours) | **50.1** | 39.1 | **53.1** |

Table 3: Ablation studies by overall test accuracy (%). The last line represents LANO's components. The best results for each dataset column are highlighted in bold and pink.

| Components | | | | | | Citeseer | | | Coauthor_CS | | |
|---|---|---|---|---|---|---|---|---|---|---|---|
| LLMs | Projector | Influence | Uncertainty | Variant1 | Variant2 | all | seen | novel | all | seen | novel |
| − | ✓ | ✓ | ✓ | − | − | 68.7 | 71.7 | 64.0 | 77.4 | 83.5 | 71.0 |
| ✓ | − | ✓ | ✓ | − | − | 69.5 | 71.7 | 66.2 | 82.6 | **85.9** | 78.4 |
| ✓ | ✓ | − | ✓ | − | − | 69.1 | **74.8** | 63.7 | 82.7 | 82.3 | **81.8** |
| ✓ | ✓ | ✓ | − | − | − | 68.2 | 66.2 | **69.4** | 80.1 | 79.8 | 81.6 |
| ✓ | ✓ | ✓ | ✓ | ✓ | − | 67.1 | 69.3 | 65.7 | 68.7 | 70.8 | 67.5 |
| ✓ | ✓ | ✓ | ✓ | − | ✓ | 67.3 | 68.7 | 66.3 | 81.4 | 82.2 | 80.0 |
| ✓ | ✓ | ✓ | ✓ | (Our Full LANO) | | **70.2** | 73.8 | 66.2 | **83.4** | 85.2 | 80.7 |

## 4.3 ABLATION STUDIES

We further conduct ablation experiments to investigate the contribution of each component of our framework. Specifically, we design six variants: 1) Removing LLM-assisted pseudo-label generation; 2) Treating projection heads as fixed rather than learnable parameters; 3) Omitting the maximum activation criterion (influence) in high-value node selection; 4) Omitting the uncertainty measure in high-value node selection; 5) Replacing informed selection with random node sampling (Variant1); 6) Replacing the multi-head architecture with a single head (Variant2). Based on these six variants, we perform repeated experiments on the Citeseer and Coauthor_CS datasets, with the results summarized in Table 3. The experiments reveal that excluding LLM-assisted pseudo-label generation leads to a substantial drop in overall classification accuracy. A further breakdown between seen and unseen classes shows that performance on seen classes remains largely unaffected, whereas the accuracy on unseen classes degrades significantly. This highlights the critical role of the LLM in enhancing the recognition of unseen classes under the open-world setting. Regarding high-value node selection, removing the uncertainty measure causes a larger performance decline compared to removing the maximum activation criterion, suggesting that uncertainty is more effective in identifying valuable unlabeled nodes. Moreover, any metric-based selection strategy yields a clear advantage over random sampling. For the multi-head design, replacing it with a single-head structure results in performance degradation, as the absence of multi-view semantic information hampers the model's ability to learn well-defined decision boundaries.

## 4.4 HYPER-PARAMETERS SENSITIVITY ANALYSIS

We further investigate the impact of four key hyperparameters on the performance of the proposed method by adopting a single-factor control strategy, i.e., when analyzing one hyperparameter, the others are fixed at their optimal values.

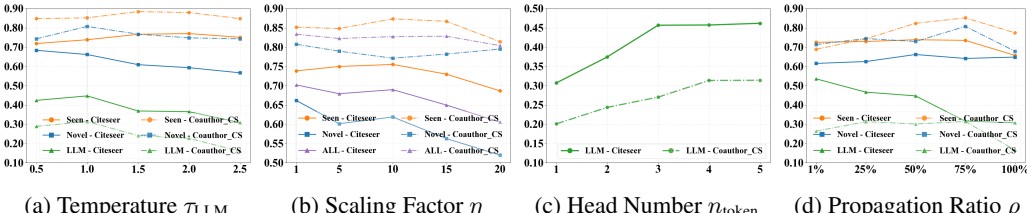

(a) Temperature $\tau_{\text{LLM}}$     (b) Scaling Factor $\eta$     (c) Head Number $n_{\text{token}}$     (d) Propagation Ratio $\rho$

Figure 3: Hyper-parameter sensitivity analysis on Citeseer and Coauthor_CS datasets. The vertical axis represents accuracy. The green LLM curve indicates the performance obtained when the LLM serves as the labeling source.

**The temperature $\tau_{\text{LLM}}$.** Figure 3a illustrates the effect of $\tau_{\text{LLM}}$ on the classification accuracy of seen classes, unseen classes, and LLM annotation. The overall trend exhibits an **increase-then-decrease** pattern: as $\tau_{\text{LLM}}$ increases (i.e., the similarity computation becomes more relaxed), there exists an optimal point for distinguishing between same-class and different-class nodes. At moderate values, $\tau_{\text{LLM}}$ effectively pulls together nodes of the same class while pushing apart nodes of different classes, thereby enhancing discriminative capability. However, when $\tau_{\text{LLM}}$ becomes too large, the similarity distribution is overly smoothed, weakening the constraints of contrastive learning. Consequently, the model's discriminative power decreases, and both the classification accuracy for seen and unseen classes as well as the reliability of LLM-generated pseudo-labels decline significantly.

**The scaling factor $\eta$.** Figure 3b illustrates the impact of scaling factor $\eta$ in the Equation 14 on the accuracy of LLM annotations. On Citeseer, the accuracy of seen classes shows a typical "rise-then-fall" trend as scale increases: small scale yields insufficient feature contrast; moderate scale enhances inter-class separability; overly large scale causes overfitting to seen classes. For novel classes, accuracy decreases overall with increasing scale, consistent with OpenIMA's findings that stronger supervision biases the model toward seen classes. The slight rebound near scale = 10 likely comes from clearer seen-class decision boundaries, which indirectly benefit novel-class separability. On Coauthor_CS, seen classes follow the same "rise-then-fall" pattern. The novel classes, however, show a "fall-then-rise" trend. We attribute this to the following: small scale leads to weak supervised learning but still allows contrast-learning coarse boundary learning, giving relatively high novel accuracy; increasing scale strengthens the bias toward seen classes, reducing novel performance; when scale becomes very large, seen classes overfit, while novel classes—guided only by LLM pseudo-labels—benefit from stronger feature stretching and thus recover in accuracy.

**The number of heads $n_{\text{token}}$.** Figure 3c illustrates the impact of $n_{\text{token}}$ on the accuracy of LLM annotations. The results show that as $n_{\text{token}}$ increases, the accuracy of LLM annotations gradually improves. However, performance bottlenecks are observed on both *citeseer* and *coauthor_cs*: on *citeseer*, the optimal performance is achieved with approximately 3 tokens, whereas on *coauthor_cs*, a larger number of tokens is required to sufficiently capture semantic information. This discrepancy may be attributed to the smaller number of classes and nodes in *citeseer*, where fewer tokens are sufficient to cover node features, thus reaching the performance optimum earlier.

**The pseudo label selection rate $\rho$.** Figure 3d illustrates the impact of the pseudo-label ratio $\rho$ on performance. On both *citeseer* and *coauthor_cs*, the accuracies of seen classes, unseen classes, and LLM annotations all achieve their optimum when $\rho$ is set to 50% or 75%. However, as $\rho$ continues to increase, the performance degrades, with the accuracy of LLM annotations exhibiting a cliff-like drop. We attribute this phenomenon to the fact that an excessively large $\rho$ leads the model to overconfidently assign incorrect pseudo-labels to nodes, which disrupts feature encoding, blurs the decision boundary, and directly results in a drastic decline in the reliability of LLM annotations.

### 4.5 VISUALIZATION

To further illustrate the effectiveness of our model, we present the visualization results of the proposed method compared with OpenIMA, along with the corresponding predicted categories and ground-truth labels in Figure 4. On the Citeseer dataset, the node distribution produced by Open-IMA is relatively scattered. While it reveals some degree of class separation, the clusters are neither compact nor well-defined. On the Coauthor_CS dataset, OpenIMA yields comparatively tighter

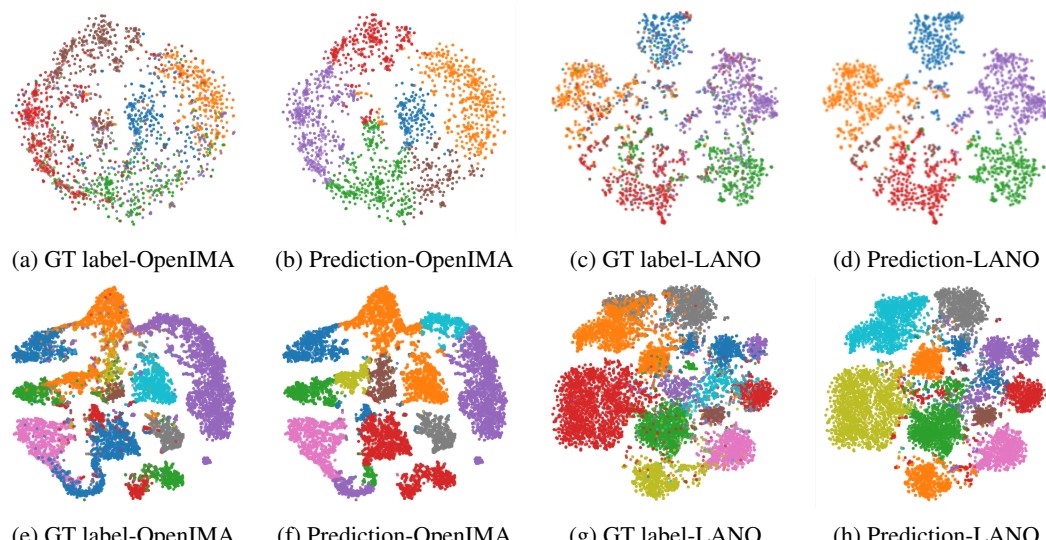

(a) GT label-OpenIMA  (b) Prediction-OpenIMA  (c) GT label-LANO  (d) Prediction-LANO

(e) GT label-OpenIMA  (f) Prediction-OpenIMA  (g) GT label-LANO  (h) Prediction-LANO

Figure 4: The T-SNE visualizations of target node representations for the Citeseer ((a)-(d)) and Coauthor_CS ((e)-(h)) datasets, where different colors indicate different classes. Because the feature spaces produced by different methods (OpenIMA vs. LANO) differ significantly in both dimension and distribution, the corresponding ground-truth (GT) label distributions ((a) and (c)) must be visualized separately to accurately assess clustering consistency and alignment with true labels.

clusters, yet the overall cluster boundaries remain indistinct. We attribute this to its pseudo-labeling strategy, which assigns labels only to the top-$\rho\%$ of nodes closest to cluster centroids. When decision boundaries are ambiguous, this approach tends to cause oscillations in pseudo-label assignments for the same node, leading to unstable supervision signals and consequently undermining both discriminative power and clustering quality. In contrast, our method integrates a multi-head architecture with LLM-assisted pseudo-label generation, further enhanced by soft labeling and label-smoothing updates. The visualizations clearly demonstrate that, across both datasets, nodes of the same class form compact and well-delineated clusters. These results suggest that our approach is capable of learning higher-quality and more discriminative graph representations.

## 5 CONCLUSION

In this work, we address the challenge of node classification under the **open-world** setting, where novel classes inevitably emerge beyond the scope of training labels. To tackle the limitations of conventional GNN-based approaches under the closed-world setting, we propose **LANO**, a novel framework that leverages LLMs, which exhibit remarkable zero-shot ability, as active annotation agents for open-world node classification. Specifically, LANO aligns GNN representations with LLM token embeddings through instance- and feature-aware self-supervised learning, enabling LLMs to serve as zero-shot predictors for graph tasks. An influence- and uncertainty-driven node selection strategy is introduced to identify representative samples for annotation, while the soft feedback propagation mechanism and bias-reduced pseudo label concordance effectively suppress the risk of propagating noisy labels and mitigate labeling bias, incorporating bias-reduced pseudo labels into iterative GNN training. Extensive experiments on multiple baselines demonstrate the effectiveness of LANO, highlighting the potential of integrating LLMs with GNNs for open-world graph learning.

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

# A  RELATED WORK

## A.1  OPEN-WORLD SEMI-SUPERVISED GRAPH LEARNING

Open-world semi-supervised learning (OW-SSL) (Cao et al., 2021) on graphs aims to address the realistic scenario where unlabeled test nodes may belong to novel classes that are unseen during training. A representative method, ORCA (Cao et al., 2021), originally proposed in the vision domain, introduces an uncertainty-based adaptive margin to balance intra-class variance. Within an end-to-end framework, it jointly tackles both classification and clustering, enabling the discovery of novel categories in an open-world setting. More recently, OpenIMA (Wang et al., 2024b) has explored a more practical OW-SSL setting by introducing a two-stage training framework. Through contrastive learning and bias-reduced pseudo-labeling, OpenIMA mitigates the imbalance between seen and novel classes and improves classification accuracy. Despite these advances, existing OW-SSL approaches still face key challenges: (1) they often rely on extensive human annotations for initial supervision, (2) their performance can degrade under distributional shifts.

## A.2  LARGE LANGUAGE MODELS FOR GRAPH LEARNING

Although LLMs are not inherently designed to capture relational structures in graphs (Wang et al., 2024a), recent studies have begun to explore their potential in graph reasoning tasks. The performance of such methods, however, strongly depends on graph encoding strategies, prompt engineering, and the structural properties of the input graph (Fatemi et al., 2023). For example, ALUP (Liang et al., 2024) investigates generalized category discovery (GCD) by integrating LLMs with active learning strategies, thereby improving the recognition of novel classes while maintaining reliable feedback and significantly reducing annotation costs. Relevant LLMs-as-graph-annotators methods, such as LLMNodeBed (Wu et al., 2025) and LLM-GNN (Chen et al., 2023) assume a closed-world setup, which makes conventional "LLM → pseudo labels → GNN training" pipelines highly error-prone, as LLM outputs can be uncertain, biased, or may indicate new classes. In summary, while open-world graph learning methods focus on OOD detection and novel class discovery, they remain constrained by annotation costs and distributional shifts. LLM-based approaches, on the other hand, offer strong generalization and zero-shot capabilities but suffer from structural limitations, annotation noise, and dependency on encoding strategies. These gaps motivate the design of a hybrid framework that integrates GNN-based representation learning with LLM-driven annotation to advance open-world node classification.

# B  DATASETS

This section provides a more detailed introduction to the popular datasets commonly used for node classification, including Citeseer, Coauthor_CS and Coauthor_Physics, Amazon_Photos, and Amazon_Computers, as shown in Table 4.

In our experimental setup, we randomly select 50% of the classes in each dataset as known classes, while the remaining classes are treated as unknown classes. For each known class, 50 nodes are randomly sampled as the training set, another 50 nodes as the validation set, and the rest are used for testing.

Table 4: Statistics of Datasets

| Graph | Type | Nodes | Edges | Features | Class |
|---|---|---|---|---|---|
| Citeseer | Citation network | 3327 | 4277 | 3703 | 6 |
| Coauthor CS | Co-author network | 18333 | 81894 | 6805 | 15 |
| Coauthor Physics | Co-author network | 34493 | 247962 | 8415 | 5 |
| Amazon Photos | Amazon network | 7650 | 119082 | 745 | 8 |
| Amazon Computers | Amazon network | 13752 | 245861 | 767 | 10 |
| ognb-Arxiv | Citation network | 169343 | 1166243 | 128 | 40 |

Table 5: Summary of important notations with their definitions.

| Symbol | Definition |
|---|---|
| $\mathcal{G} = (\mathcal{V}, \mathcal{E}, \mathbf{X}, \mathbf{A})$ | Graph with node set $\mathcal{V}$, edge set $\mathcal{E}$, feature matrix $\mathbf{X}$, and adjacency matrix $\mathbf{A}$. |
| $N = |\mathcal{V}|$ | Total number of nodes |
| $\mathcal{V}_l, \mathcal{V}_u$ | Set of labeled nodes / Set of unlabeled nodes |
| $\mathcal{C}_l, \mathcal{C}_u$ | Set of labeled (seen) classes / Set of unlabeled classes |
| $\mathcal{C}_n$ | Set of novel classes, $\mathcal{C}_n = \mathcal{C}_u \setminus \mathcal{C}_l$ |
| $\mathcal{Y}_l, \mathcal{Y}_u$ | Set of labels for nodes in $\mathcal{V}_l, \mathcal{V}_u$ |
| $\mathcal{N}(v_i)$ | Neighbor set of node $v_i$ |
| $U_*$ | Node representations encoded by the GNN; $* \in \{1, 2\}$ denotes different views. |
| $\boldsymbol{w}_i, \boldsymbol{w}_i'$ | Embeddings of node $i$ from the two views, used for contrastive learning. |
| $\tau$ | Temperature parameter |
| $\boldsymbol{m}_i, \boldsymbol{n}_i$ | Feature vectors from augmented embeddings |
| $u(v_i)$ | Uncertainty of node $v_i$ |
| $r_{v_i}$ | Influence quality score of node $v_i$ |
| $Q(v_j, v_i, k)$ | Influence of node $v_i$ on node $v_j$ after $k$ propagation steps |
| $\text{score}(v_i)$ | Node selection score |
| $\mathcal{L}$ | A loss function |
| $y_i^{\text{LLM}}$ | Pseudo-label generated by the LLM for node $v_i$ |
| $S$ | Set of selected nodes for LLM annotation |
| $y_j^{\text{prop}}$ | Propagated pseudo-label for node $v_j$ |
| $\hat{\mathcal{Y}}^t$ | Aggregated pseudo-label matrix at iteration $t$ |
| $\hat{\mathcal{Y}}_{\text{new}}^t$ | Newly generated pseudo-label matrix at iteration $t$ |
| $\gamma$ | Label decay coefficient |
| $\eta$ | Scaling factor for cross-entropy loss $\mathcal{L}_{\text{CE}}$ |

## C  IMPORTANT NOTATION TABLE

In this section, we summarize the important notations and their definitions, as shown in Table 5:

## D  PROMPTS

In this part, we show the motivation and how to construct the prompt for our task.

**Motivation.** Because LLMs do not naturally understand graph structures or node features, enabling the LLM to act as an annotator requires addressing two key challenges: (1) Heterogeneous space alignment (GNN representations → LLM token embeddings), and (2) Label reliability: LLM-generated pseudo-labels are susceptible to hallucinations and biases, requiring mechanisms to improve classification quality and annotation stability.

**Construction.** Our prompt consists of four detailed parts: (1) Task Instruction (defining open-world node classification questions), (2) Category Description, (3) Node Representation Tokens, and (4) Output Rules.

- In the **Category Description** section, LANO proposes a comparison-based prompting method, where the LLM compares a sample with representative samples of different classes to improve classification quality.

- In the **Node Representation Tokens** section, LANO tackles the alignment problem by projecting graph representations into a fixed number of Graph Token Embeddings, inserted into a graph slot for LLM parsing, with a unified strategy across tasks (e.g., using center-node embeddings for node tasks).

- Our LANO adopts structured **output rules** (label + confidence level), which support the recognition of Known / New / Uncertain categories, to improve annotation stability.

**Prompt.** The complete prompt is shown as follows:

---

**Prompt for Large Language Model Annotation in Open-World Node Classification**

Your task is to classify the given target node into one of the predefined categories or determine if it belongs to a new category based on semantic similarity.
Please carefully read the following set of predefined categories, where each category is represented by a list of representative semantic tokens:

```
 {% for category in categories -%}
category {{ category.id }}:
{% for token in category.tokens%}{{token}}{% endfor %};
{% endfor % }
```

Now, here is the target node represented by a token sequence:
```
 {% for token in target_tokens %}{{token}}{% endfor %};
```

**When classifying the target node, please follow these rules:**
1. **Match to Known Category:**
If the target node's semantics strongly align with one of the known category $X$ (where $X$ is a number), output the result in the format:
`[X][ConfidenceLevel]`
Confidence Levels:
`A`: $\geq 99\%$ confidence
`B`: $\geq 75\%$ confidence
`C`: $\geq 50\%$ confidence
`D`: $\geq 25\%$ confidence

2. **New Category Detection:**
If the target node's semantics are inconsistent with all known categories but indicate a novel and coherent class, return:
`[N][ConfidenceLevel]`

3. **Uncertain Classification:**
If the classification is ambiguous or insufficient evidence is available, return:
`[-1]`

Please only return the final classification label (e.g.`[X][ConfidenceLevel]`, `[N][ConfidenceLevel]`, `[-1]`. $X$ represents a number, do not output $X$). Do not output explanations or reasoning.

---

## E  EVALUATION METRIC

**LLM Labeling Accuracy.** During the process of generating pseudo-labels with LLMs, we evaluate their consistency with the ground-truth labels. According to our prompt design rules, if a node is classified by the LLM as belonging to a visible class, the pseudo-label is considered correct only if it matches the ground truth. If a node is classified as belonging to an unseen class, we instead refer to the output of the classification head as its pseudo-label, and correctness is determined by whether this label matches the ground truth. If the LLM outputs "unrecognizable," the labeling result is discarded and excluded from evaluation. All other cases—including assigning incorrect labels or producing non-standardized outputs—are regarded as labeling errors.

## F  BASELINES

This section introduces the baselines used for comparison with our method in the experiments, as summarized in Table 1. The details are as follows:

- **OODGAT**: Out-of-Distribution Graph Attention Network (OODGAT) is a GNN model that explicitly models the interactions among different types of nodes and separates in-distribution and out-of-distribution nodes during feature propagation.

- **OpenWGL**: A new paradigm for open-world graph learning, whose objective is not only to classify nodes belonging to visible classes correctly but also to classify nodes outside the known classes into the unseen category.

- **ORCA**: A model specifically designed to address open-set recognition under long-tailed distributions. Its main objective is to prevent the model from misclassifying rare unknown samples as rare known classes. ORCA without the margin mechanism is denoted as ORCA-ZM.

- **SimGCD**: A simple yet effective method for generalized category discovery (GCD), aiming to simultaneously recognize known classes and discover unknown classes.

- **OpenLDN**: OpenLDN employs a pairwise similarity loss to discover novel classes. Leveraging a bi-level optimization scheme, the pairwise similarity loss exploits available information from the labeled set to implicitly cluster samples of new categories while identifying samples from known categories.

- **OpenCon**: A method targeting open-world classification and novel class clustering by combining semi-supervised learning with contrastive losses. It determines the total number of classes, introduces a classification head, and optimizes using contrastive objectives tailored for labeled, unlabeled, and novel categories.

- **InfoNCE**: Noise Contrastive Estimation Loss is a widely used self-supervised loss function for representation learning. Rooted in information-theoretic principles, it learns model parameters by contrasting the similarity between positive and negative samples.

- **OpenIMA**: OpenIMA is designed for open-world semi-supervised node classification. It trains a classifier from scratch using unbiased pseudo-labels and contrastive learning, effectively mitigating intra-class imbalance and improving classification accuracy. Compared to many existing node classification approaches, OpenIMA demonstrates superior performance.

## G    IMPLEMENTATION DETAILS

**Design of GNN Heads.**    All GNN heads adopt Graph Attention Networks (GAT) as the feature encoder. To enable multi-perspective node representation, we vary the hop number, hidden dimension, number of attention heads, dropout rate, and whether to apply residual connections across different heads. For compatibility with the projector, the output dimension of the final GAT layer is fixed to 256. During training, we employ the Adam optimizer with a weight decay of $1 \times 10^{-4}$. Since both feature and attention mechanisms involve dropout, we sample each input twice to construct positive pairs for contrastive learning.

**Obtaining Low-dimensional LLM Embeddings.**    To efficiently leverage the semantic representations of LLMs under limited GPU memory, we perform dimensionality reduction on their high-dimensional embeddings. As the datasets used lack bag-of-words information, semantic textual features cannot be directly obtained. Instead, we encode node attributes as tokens and feed them into the LLM to obtain semantic embeddings. Taking QWen3-8B as an example, the embedding dimension typically exceeds 4000, which would incur prohibitive GPU memory costs if directly used for supervised contrastive loss. Therefore, we apply PCA to reduce embeddings to 1000 dimensions. The projection head then maps generated word vectors into this reduced space, where supervised contrastive loss is computed for semantic alignment.

**Choice of LLM.**    Prior studies suggest that LLM performance saturates around 7B parameters, with marginal gains from larger models. Considering model scale and release time, we evaluate Mistral-7B, Qwen3-8B, and Deepseek-r1-7B. Experiments show that Mistral-7B often outputs "uncertain" predictions with very low confidence, while Deepseek-r1-7B, due to its built-in chain-of-thought mechanism, generates unnecessarily long reasoning when fed non-standard token embeddings, leading to high inference cost. In contrast, Qwen3-8B provides stable outputs with controllable reasoning and more efficient inference. Consequently, we adopt Qwen3-8B as the pseudo-label generator in subsequent experiments.

**Soft Pseudo-labels from LLM.**  Since the input tokens fed into the LLM are multi-head encoded rather than complete natural language text, directly producing hard labels may introduce bias. To address this, we design the LLM outputs as "label + confidence level," a soft pseudo-label format analogous to a softmax distribution. This not only reflects predictive uncertainty but also better supports pseudo-label propagation and forgetting mechanisms, thereby improving overall robustness.

**Experimental Hyperparameter Settings.**  When inheriting default hyperparameters from the framework, we set the scaling factor $\eta = 1$, temperature $\tau = 0.7$, and pseudo-label selection rate $\rho = 75\%$. An additional supervised contrastive loss is introduced for low-dimensional projection alignment. To prevent unstable training caused by large gradients, we apply a scaling factor of 0.025 to the GAT embeddings. Parameter search shows that as model complexity increases, stronger performance on the *amazon_computers* and *amazon_photos* datasets requires increasing $\eta$ to 30 and slightly lowering $\tau$, making the model more confident in pseudo-labeling while slowing down the forgetting rate to retain high-confidence pseudo-labels. In contrast, for *citeseer*, where validation accuracy is significantly lower, we adopt a stricter strategy by reducing the pseudo-label selection rate $\rho$ and maintaining a higher forgetting rate, so as to mitigate the adverse impact of noisy pseudo-labels in early training.

## H  CALCULATION OF INFLUENCE QUALITY

The influence quality $r_{v_i}$ in equation 8 is not a predefined constant; it is dynamically updated in each propagation round based on the current node features and oracle accuracy. As training progresses, node representations and semantic similarities evolve, enabling adaptive estimation of node influence quality. Specifically, we introduce a dynamic feature similarity and confidence-adjusted mechanism to achieve adaptive computation. We first construct the local propagation range based on the graph's two-hop adjacency and compute the semantic similarity between nodes:

$$q_{ij} = \text{cosine}(p_i^{(2)}, p_j^{(2)}) \tag{15}$$

where $p_i^{(2)}$ denotes the two-hop aggregated feature of node $v_i$. Then, integrating structural similarity and label confidence, the influence quality between nodes is defined as:

$$r_{ij} = \frac{\alpha_i q_{ij}}{\alpha_i q_{ij} + (1 - \alpha_i)\frac{1 - q_{ij}}{C_N - 1}} \tag{16}$$

where $C_N$ is the number of classes and $\alpha_i$ reflects the confidence weight of the LLM annotation. The overall influence quality of a node $v_i$ is given by the average reliable influence over its two-hop neighborhood:

$$r_{v_i} = \frac{1}{|\mathcal{N}_2(v_i)|} \sum_{v_j \in \mathcal{N}_2(v_i)} \mathbf{1}\big[(\mathbf{A}_{ij}^2 \cdot r_{ij}) > \theta\big] \tag{17}$$

where $\mathbf{A}_{ij}^2$ denotes the two-hop adjacency matrix, $\theta$ is the activation threshold, and $\mathbf{1}$ is an indicator function.

## I  COST ANALYSIS OF LLMS

In this section, we analyse the cost of using the LLM explicit, including the number of calls, average prompt length, and wall clock per dataset, to would make the method much easier to adopt. We adopt a locally deployed Qwen3-8B model running on a single RTX 4090 GPU. For each dataset, we run the full pipeline three times consecutively and report the approximate statistics (average prompt length and wall clock) in Table 6.

**Number of calls.** In the early stages of training, the model has not yet learned strong encoding abilities, and the token representations still contain substantial noise. As a result, the LLM often outputs "uncertain" when queried, leading to unnecessary computational overhead.

Therefore, we introduce LLM-based labeling only in the later stages of training. In each epoch, we query the LLM with 10 selected nodes. A typical training process contains around 40 epochs, so an entire experiment involves approximately 200 queries. However, we emphasize that this fixed number of epochs is used solely to verify that the training is sufficiently converged. If early stopping is enabled, the total number of queries could be significantly smaller.

Table 6: The LLM usage cost of LANO, including average prompt length (tokens) and wall clock (HH:MM:SS).

| Dataset | Avg Prompt Length (tokens) | Avg Wall Clock Time |
|---|---|---|
| Coauthor_CS | 383 | 00:16:33 |
| Coauthor_phy | 372 | 03:14:07 |
| Citeseer | 328 | 00:11:01 |
| Amazon_photos | 339 | 00:10:41 |
| Amazon_computers | 350 | 00:21:01 |
| ogbn-arxiv | 570 | 04:48:44 |

These results indicate that our approach remains computationally feasible even with a locally running 8B model.

## J    CONFIDENCE-BASED ACCURACY ANALYSIS AND "UNDECIDABLE" RATE

We apply confidence-based thresholding to filter the LLM's parsed outputs, using three thresholds: 50%, 75%, and 99%. For each threshold, we compute the LLM's labeling accuracy using only the predictions whose confidence exceeds the corresponding cutoff. As shown in the Table 7, the labeling accuracy consistently increases as the confidence threshold becomes higher, indicating that our threshold buckets effectively encourage the LLM to produce more reliable and robust predictions.

**"Undecidable" Rate**: In our experiments, we did not explicitly log the number of "unable to determine" responses. However, based on available logs and backward estimation from the total annotations, we find that—with appropriate parameter configurations—the model returns an "unable to determine" label in approximately 25%–40% of cases, depending on the dataset. This behavior is consistent with our design: the LLM refrains from assigning a label when its confidence score falls below the threshold, thereby maintaining the reliability of the generated annotations.

Table 7: Annotation accuracy (%) of the LLM across different datasets at different thresholds (50%, 75%, and 99%).

| Dataset | 50% | 75% | 99% |
|---|---|---|---|
| Citeseer | 46.79 | 48.98 | 51.88 |
| Coauthor_CS | 31.58 | 33.33 | 35.71 |
| Amazon_photos | 27.91 | 33.14 | 38.04 |
| Amazon_computers | 23.92 | 32.91 | 36.28 |

## K    COMPARISON WITH LLM-BASED GRAPH METHODS

We compare experiments with the most relevant LLMs-as-graph-annotators method, LLM-GNN (Chen et al., 2023). Although LLM-GNN and LANO both involve LLMs, their settings differ fundamentally: (1) LANO targets **open-world** node classification, requiring detection and handling of novel classes, whereas LLM-GNN assumes a **closed-world** setup; (2) LLM-GNN operates on **Text-Attributed Graphs (TAGs),** inserting raw node text directly into prompts with limited use of structure, while LANO does not rely on textual attributes—using node feature vectors, GNN-based encoding, and explicit GNN–LLM representation alignment; (3) LLM-GNN follows a label-free zero-shot paradigm, whereas LANO performs **open-world semi-supervised learning** with a labeled–unlabeled split, making direct comparison inappropriate.

To compare the newest lines of work that also use LLMs as graph annotators or weak oracles with our LANO, we adapted LLM-GNN, which is originally designed under a closed-world zero-shot setting, to the open-world semi-supervised setting to ensure a fair comparison. Specifically, we preset 50

labeled nodes, exposing only half of the classes as seen and treating the remaining classes as unseen. We then followed the LLM-GNN training pipeline—difficulty-aware active node selection, confidence-aware annotation, optional post-filtering, and GNN training & prediction. The final performance on the Citeseer dataset under this adapted setting is reported in Table 8. Based on the above experimental results and our ablation experiments, we confirm: (1) Most of LANO's performance gain does not simply come from using an LLM. (2) Our active selection + bias-controlled propagation are the key contributors.

Table 8: Comparison of LANO with other relevant work that also use LLMs as graph annotators, such as LLM-GNN, on CiteSeer dataset. The best results in each column are highlighted in bold.

| Method | Citeseer | | |
|---|---|---|---|
| | all | seen | novel |
| OpenIMA | 68.1 | 71.8 | 64.3 |
| LLM-GNN | 66.9 | 66.3 | **67.5** |
| **LANO (Ours)** | **70.2** | **73.8** | 66.2 |

## L  LARGE LANGUAGE MODELS USAGE STATEMENT

In the preparation of this research, large language models (LLMs) were employed strictly as a limited-purpose auxiliary tool. The models were used exclusively for language polishing tasks, including grammar checking, sentence structure optimization, and wording refinement to improve the readability and linguistic fluency of portions of the text. The LLMs played no role in any core research activities, including but not limited to: research ideation, theoretical development, experimental design, data analysis, result interpretation, or scientific decision-making. All intellectual contributions to this work originate solely from the human authors. The authors take full responsibility for the entire content of this paper, including text polished by LLMs, and affirm its originality, accuracy, and academic integrity.

