# OpenReview forum: "LANO: Large Language Models as Active Annotation Agents for Open-World Node Classification"
_ICLR.cc/2026/Conference — Submitted to ICLR 2026_

### Official Review · Reviewer_cF53 · 2025-10-22

**Soundness:** 2
**Presentation:** 2
**Contribution:** 2
**Rating:** 4
**Confidence:** 3

**Summary:**

This paper proposes LANO, a framework that leverages LLM as active annotation agents for open-world semi-supervised node classification. The method aims to reduce reliance on expensive, high-quality predefined labels by aligning GNN and LLM representations, selecting informative nodes for LLM annotation, and propagating bias-reduced pseudo-labels for iterative GNN training. Experimental results show performance gains over baselines, suggesting that the approach positively impacts open-world scenarios.

**Strengths:**

1. The idea of using LLMs as active annotation agents for graph learning is timely and relevant, especially for open-world (novel/unseen class) scenarios.
2. The paper addresses a real deployment issue (label scarcity and emergent classes), and therefore has clear potential practical value.

**Weaknesses:**

1. Limited novelty and insufficient experiments.
Using LLMs for label generation or annotation assistance has already been explored in prior work. The paper does not clearly articulate what is fundamentally new compared with existing LLM-assisted methods. Conversely, if the main contribution lies in proposing a new framework for LLM utilization, the paper fails to position LANO precisely within related work.
In particular, the only baseline table (from OpenIMA) is not LLM-based. A fair comparison should incorporate LLM-GNN baselines and ablations over different LLM choices, similar to [1].

2. Paper quality and presentation.
The paper contains numerous typographical errors, inconsistent notations, and several malformed or ambiguous equations (including but not limited to detailed list below). These issues significantly reduce readability and raise concerns about correctness and reproducibility.
(1) Line 041: “... high-quality labeled, ...” -> should be “high-quality labeled data, ...”
(2) Line 126: The explanation for statement $C_l \cap C_u \neq \varnothing$ is misleading. In the source paper [2], $C_u$​ denotes "the set of classes associated with the nodes in $V_u$", which is similar but completely different from "the classes of unlabeled nodes" as described here. The explanatory sentences are therefore incorrect and must be rewritten to match the formal definitions.
(3) Line 127: The adjacency matrix A is referenced, but the graph definition $G=(V,E,X)$ in line 105 does not include A. The usage of A (e.g., how $\tilde{A}$ is perturbed), probably taken from [3], is also missing. This inconsistency appears inherited from the differnet source papers and not properly corrected.
(4) Line 133: The subscript $u$ is reused in Eq. (1), where it previously denoted the "unlabeled set". Reusing the same symbol for different meanings causes confusion.
(5) Line 183: The formatting of Eq. (4) is corrupted.
(6) Line 188: Ungrammatical sentence: “To bridge the gap ..., the goal is to ensure...”
(7) Figure 4: Lacks clarification and is difficult to interpret. The comparison with different ground truths also appears unfair.

[1] Wu X, Shen Y, Ge F, et al. When Do LLMs Help With Node Classification? A Comprehensive Analysis[J]. arXiv preprint arXiv:2502.00829, 2025.
[2] Wang Y, Zhang J, Zhang L, et al. Open-world semi-supervised learning for node classification[C]//2024 IEEE 40th International Conference on Data Engineering (ICDE). IEEE, 2024: 2723-2736.
[3] Wang D, Zuo Y, Li F, et al. Llms as zero-shot graph learners: Alignment of gnn representations with llm token embeddings[J]. Advances in Neural Information Processing Systems, 2024, 37: 5950-5973.

**Questions:**

1. “Two views” in Fig. 2 are not explained in the text. Please clarify what the perturbed views refer to and how they function.
2. Eqs. (4) and (5) appear to be modified from (Wang et al. 2024a) with parts of the denominator removed. Why were those terms deleted, as this might affect numerical stability and the behavior of contrastive learning?
3. Figure 4 inconsistency. The figure appears to depict a 2×3 grid (two datasets × three label variants: GT / OpenIMA / LANO) but actually contains eight subplots. Why are there duplicated ground-truth plots for the same dataset?

---

> ### Author Response · Authors · 2025-11-21
>
> We are truly thankful for your insightful and constructive review. Our detailed responses are presented below.
> ***
> ## W1: Limited novelty and insufficient experiments
>
> **A1**: We thank the reviewer for this thoughtful comment. While prior work such as LLMNodeBed\[1] also leverages LLMs for node classification, **LANO** targets a fundamentally different setting—**open-world node classification**—which must distinguish seen classes, detect unseen classes, and support novel-class discovery. This setting makes conventional “LLM → pseudo labels → GNN training” pipelines highly error-prone, as LLM outputs can be uncertain, biased, or indicate new classes. Compared with LANO, prior works assume a **closed-world** setup.
>
> **Methodologically**, LANO introduces graph self-supervised alignment, which aligns the GNN representations with LLM token embeddings, to enable structure-aware zero-shot reasoning **beyond text-attributed graphs(TAG)**, an influence- and uncertainty-based node selection strategy to identify the most informative nodes, and a bias-reduced soft propagation mechanism that uses only high-confidence, neighbor-consistent pseudo labels to avoid error reinforcement under open-world conditions. Thus, direct comparison between LLMNodeBed\[1] and our LANO across such distinct settings would therefore be unfair and potentially misleading.
>
> **Additional Experiments**: To address the reviewer’s concern, we compare experiments with the most relevant LLMs-as-graph-annotators method, **LLM-GNN**[2]. Since LLM-GNN is originally designed under a **closed-world zero-shot** setting, we adapted it to the **open-world semi-supervised** setting to ensure a fair comparison. Specifically, we preset 50 labeled nodes, exposing only half of the classes as **seen** and treating the remaining classes as **unseen**. We then followed the LLM-GNN training pipeline—difficulty-aware active node selection, confidence-aware annotation, optional post-filtering, and GNN training & prediction.
>
> The final performance on the Citeseer dataset under this adapted setting is reported in Table 1.
>
> > Table 1: Comparison of LANO with other relevant work that also use LLMs as graph annotators, such as LLM-GNN, on CiteSeer dataset. The best results in each column are highlighted in bold.
>
> | Method  | all      | seen     | novel    |
> | ------- | -------- | -------- | -------- |
> | OpenIMA | 68.1     | 71.8     | 64.3     |
> | LLM-GNN | 66.9     | 66.3     | **67.5** |
> | LANO    | **70.2** | **73.8** | 66.2     |
>
> Based on the above experimental results and our ablation experiments, we confirm: (1) Most of LANO’s performance gain does not simply come from using an LLM. (2) Our active selection + bias-controlled propagation are the key contributors.
>
> **Different LLM choices**: For the **choice of LLM**, we have shown this in Appendix F (Appendix G in the revised version). Prior studies\[1] suggest that LLM performance saturates around 7B parameters, with marginal gains from larger models. And it is suggested to choose open-source LLMs instead of close-source LLMs for most LLM-as-Predictor methods. Considering model scale and release time, we evaluate Mistral-7B, Qwen3-8B, and Deepseek-r1-7B. Our experiments show that Mistral-7B often outputs “uncertain” predictions with very low confidence, while Deepseek-r1-7B, due to its built-in chain-ofthought mechanism, generates unnecessarily long reasoning when fed non-standard token embeddings, leading to high inference cost. In contrast, Qwen3-8B provides stable outputs with controllable reasoning and more efficient inference. Consequently, we adopt Qwen3-8B as the pseudo-label generator in subsequent experiments.&#x20;
>
> We will include these analysis in the revised manuscript. Thanks again for engaging with this point. We hope our clarification addresses your concern.

---

> ### Author Response · Authors · 2025-11-21
>
> ## W2: Paper quality and presentation
>
> **A2**: We sincerely thank the reviewer for the careful review of the manuscript and for listing specific issues. These comments are highly valuable for improving readability, formatting consistency, and symbol accuracy. We will conduct a systematic check of the manuscript and will address each point in the final version.
>
> 1. **Line 041**: “… high-quality labeled, …” → has been corrected to “high-quality labeled data, …”.
>
> 2. **Line 126**: Thank you for your careful review and correction. In the problem definition of our paper, the explanation was not fully rigorous. In the camera-ready version, we will clarify the problem definition: “...nodes from $\mathcal{C}_n$ are not only detected as unknown but further distinguished and categorized into their respective novel classes...”, where $\mathcal{C}_n$ denotes the set of novel classes and $\mathcal{C}_n=\mathcal{C}_u \setminus \mathcal{C}_l$. We will also rewrite the related paragraphs to ensure all mathematical symbols are clearly and consistently defined. Thanks again for your comment.
>
> 3. **Line 127**: In the revised version, we will standardize the notation throughout the paper and add a notation table of  all important notations and their definitions in Appendix C.
>
> 4. **Line 133**: For instance, **node embeddings** will be consistently denoted as ($\\boldsymbol{w}\_i, \boldsymbol{w}'\_i$), and $V_u$ will exclusively represent the unlabeled set. All symbols will be harmonized across the manuscript.
>
> 5. **Line 183**: Thank you for your careful review. In the camera-ready version, the Eq. (4) will be corrected as: $  \mathcal{L}\_{emb} = -\frac{1}{2N}\sum\_{i=1}^{2N}\log \frac{\exp(\text{sim}(\boldsymbol{w}\_i, \boldsymbol{w}'\_{i} / \tau))}
>  {\sum\_{j=1}^{N} \mathbf{1}\_{[j \neq i]} \exp(\text{sim}(\boldsymbol{w}\_i, \boldsymbol{w}\_{j} / \tau)) + \sum\_{j=1}^{N} \exp(sim(\boldsymbol{w}\_i, \boldsymbol{w}'\_{j} / \tau))}$, where $sim(\cdot)$ denotes cosine similarity, $\tau$ is the temperature parameter, $\boldsymbol{w}\_i, \boldsymbol{w}'\_{i}$ are embeddings of node $i$ from the two views used for contrastive learning, and $\mathbf{1}_{[j \neq i]}$ is an indicator function that takes the value 1 if $j\neq i$, and 0 otherwise.
>
> 6. **Line 188**: “To bridge the gap ..., the goal is to ensure...” will be corrected to “To bridge the gap ..., it is essential to ensure...”
>
> 7. **Figure 4**: We will add more detailed figure captions in the final version, including the rationale for visualizing two separate ground-truth distributions. Because the feature spaces produced by different methods (OpenIMA vs. LANO) differ significantly in both dimension and distribution, the corresponding ground-truth label distributions must be visualized separately to accurately assess clustering consistency and alignment with true labels.
>
> And we will correct the other typographical errors in the revised manuscript. Thanks again for your careful and valuable comments.

---

> ### Author Response · Authors · 2025-11-21
>
> ## Q1: Clarification of the "Two views" in Figure 2
>
> **A3**: We thank the reviewer for the clarification. In our experiments, the GNN is trained using **supervised contrastive learning**, which aims to **bring representations of same-class samples closer together** while **pushing apart representations of different-class samples**.
>
> For the "two views", we feed the same sample into the feature encoder twice. Due to the stochastic dropout masks applied during each forward pass, the encoder produces slightly different intermediate activations and outputs, while preserving the same underlying structural information. As a result, we obtain two distinct views derived from the same instance—forming a positive pair for the contrastive learning objective. More detailed explanation is included in the revised version.
>
> Thanks again for your comment.
> ***
> ## Q2: Modification in Contrastive Loss
>
> **A4**: We thank the reviewer for the comment. Eqs. (4) and (5) both represent the classic supervised contrastive loss function. In the revised version, we have corrected all ambiguous notations and updated Eqs. (4) to:
>
> $ \mathcal{L}\_{emb} = -\frac{1}{2N}\sum\_{i=1}^{2N}\log \frac{\exp(\text{sim}(\boldsymbol{w}\_i, \boldsymbol{w}'\_{i} / \tau))}
>  {\sum\_{j=1}^{N} \mathbf{1}\_{[j \neq i]} \exp(\text{sim}(\boldsymbol{w}\_i, \boldsymbol{w}\_{j} / \tau)) + \sum\_{j=1}^{N} \exp(sim(\boldsymbol{w}\_i, \boldsymbol{w}'\_{j} / \tau))}.$
>
> In this formulation, the denominator includes *all samples except* the anchor representation $\boldsymbol{w}_i$, covering both positive and negative samples, in order to pull together samples of the same class and push apart samples of different classes in the embedding space. Its full expanded form, which we refer to as **Eqs. (4.1)**, is:
>
> $  \mathcal{L}\_{emb} = -\frac{1}{2N}\sum\_{i=1}^{2N}\log \frac{\exp(sim(\boldsymbol{w}\_i, \boldsymbol{w}'\_{i} / \tau))}
>   {\exp(sim(\boldsymbol{w}\_i, \boldsymbol{w}'\_{i} / \tau))+\sum\_{j=1}^{N} \mathbf{1}\_{[j \neq i]} \exp(sim(\boldsymbol{w}\_i,  \boldsymbol{w}\_{j} / \tau))+\sum\_{j=1}^{N} \mathbf{1}\_{[j \neq i]} \exp(sim(\boldsymbol{w}\_i, \boldsymbol{w}'\_{j} / \tau))}.$
>
> The revised Eqs. (4.1) and Eqs. (4) express the *same mathematical meaning*, with Eqs. (4.1) providing a more explicit and readable formulation. However, Eqs. (4.1) exceeds the available space constraints, so we will adopt the more concise Eqs. (4) in the camera-ready version.
>
> Similarly, we have corrected all ambiguous symbols throughout the paper and updated Eqs. (5) to:
>
> $\mathcal{L}\_{llm} =
>     -\frac{1}{2F\_U}\sum\_{i=1}^{2F\_U}\log \frac{\exp(\text{sim}(\boldsymbol{m}\_i, \boldsymbol{n}\_i)/\tau)}
>     {\sum\_{j=1}^{F\_U} \mathbf{1}\_{[j \neq i]} \exp(\text{sim}(\boldsymbol{m}\_i, \boldsymbol{m}\_j)/\tau)+ \sum\_{j=1}^{F\_U}\exp(\text{sim}(\boldsymbol{m}\_i, \boldsymbol{n}\_j)/\tau)]}$
>
> where $\boldsymbol{m}\_i$ and $\boldsymbol{n}\_i$ are the $i$-th feature vectors from two augmented embeddings $U\_1$ and $U\_2$, and $F\_U$ is the dimension size of node representation.
>
> Thank you again for your valuable comment, and we will conduct a systematic check of the manuscript.
> ***
> ## Q3: Explanation of the Inconsistency in Figure 4
>
> **A5**: We thank the reviewer for the clarification. Figure 4 shows the feature clustering distributions of **OpenIMA** and **LANO** on two datasets, highlighting the differences between the methods.
>
> For high-dimensional feature visualization, we use **t-SNE** to project the GNN-encoded node embeddings into a two-dimensional space. However, because the feature spaces produced by different methods (OpenIMA vs. LANO) differ significantly in both dimension and distribution, the corresponding **ground-truth (GT) label distributions** must be visualized separately to accurately assess clustering consistency and alignment with true labels.
>
> Consequently, each method includes an independent **GT–Prediction** visualization, resulting in a total of 8 subplots in the figure. We will provide more detailed figure captions in the final version to improve clarity.
>
> Thanks again for pointing this out. We will clarify this in the revision.
> ***
>
> In light of these responses, we hope we have addressed your concerns, and we hope you will consider raising your score. We will properly include all the rebuttal contents in the revised version, following your valuable suggestions.
>
> ***
>
> \[1] Wu X, Shen Y, Ge F, et al. When Do LLMs Help With Node Classification? A Comprehensive Analysis\[J]. arXiv preprint arXiv:2502.00829, 2025.
>
> \[2] Zhikai Chen, Haitao Mao, Hongzhi Wen, Haoyu Han, Wei Jin, Haiyang Zhang, Hui Liu, and Jiliang
> Tang. Label-free node classification on graphs with large language models (llms). arXiv preprint
> arXiv:2310.04668, 2023.

---

### Official Review · Reviewer_QJvH · 2025-11-01

**Soundness:** 3
**Presentation:** 3
**Contribution:** 3
**Rating:** 8
**Confidence:** 5

**Summary:**

This paper leverages LLMs as zero-shot predictors for open-world node classification. The framework first aligns GNN representations with LLM token embeddings via instance-aware and feature-aware self-supervised learning. Then the paper employs influence- and uncertainty-driven strategies to select representative nodes and leverages LLMs for cost-effective pseudo-label generation. The paper also conducts extensive experiments on benchmark datasets, and the results sound the good performance of the proposed framework.

**Strengths:**

S1- The paper leverages the zero-shot capacities of LLMs for open-world node classification. The motivation is fresh.

S2- The paper aligns the GNN representations with LLM token embeddings. The alignment sounds reasonable.

S3- The results on the benchmark datasets sounds the proposed methods achieve better performance compared with baselines.

S4- The paper is well-written and easy to follow.

**Weaknesses:**

W1- The prompt for the LLM annotations if not detailed. It is suggested for better explain the motivation and how to construct the prompt for the task.

W2- It is suggested to add the scaling factor for in the equation 14 in the parameter sensitivity analysis.

W3- It is recommended to provide a table in the appendix that lists all important notations and their definitions. This would significantly improve the paper's readability and help readers better understand the equations.

W4- There are some typos. For example, in figure 3, the horizontal axis denotes LLM tau. It is suggested to replace with the notation $\tau$ for better understanding.

**Questions:**

Q1- What is the soft feedback propagation strategy, it is better to explain the strategy in a more detailed way.

Q2- How to calculate the influence quality $r_{v_i}$ in the equation 8. Is the parameter predefined of dynamic changes through the training process?

---

> ### Author Response · Authors · 2025-11-21
>
> We are truly grateful for the time you have taken to review our paper, your insightful comments, and support. Your positive feedback is incredibly encouraging for us! In the following response, we would like to address your major concern and provide additional clarification.
> ***
> ## W1: Clarifying Prompt Motivation and Construction
>
> **A1**: We thank the reviewer for the attention to the prompt design details.&#x20;
>
> **Motivation**: Because LLMs do not naturally understand graph structures or node features, enabling the LLM to act as an annotator requires addressing two key challenges: (1) **Heterogeneous space alignment** (GNN representations → LLM token embeddings), and (2) **Label reliability**: LLM-generated pseudo-labels are susceptible to hallucinations and biases, requiring mechanisms to improve classification quality and annotation stability.&#x20;
>
> **Construction**: Our prompt consists of four detailed parts: **Task Instruction** (defining open-world node classification questions), **Category Description**, **Node Representation Tokens**, and **Output Rules**.&#x20;
>
> * In the **Category Description** section, **LANO** proposes a comparison-based prompting method, where the LLM compares a sample with representative samples of different classes to improve classification quality.&#x20;
>
> * In the **Node Representation Tokens** section, **LANO** tackles the alignment problem by projecting graph representations into a fixed number of **Graph Token Embeddings**, inserted into a `<graph>` slot for LLM parsing, with a unified strategy across tasks (e.g., using center-node embeddings for node tasks).
>
> * Ou&#x72;**&#x20;LANO** adopts **structured output rules** (label + confidence level), which support the recognition of *Known / New / Uncertain* categories, to improve annotation stability.
>
> Thanks again for this valuable comment.We will include the more detailed motivation and construction of our prompt in the revised version’s Appendix C.
> ***
> ## W2: Parameter sensitivity analysis of scaling factor $\eta$
>
> **A2**: We thank the reviewer for the suggestion to include a hyperparameter sensitivity analysis for the **scale factor** $\eta$. We conducted a sensitivity analysis of the scale hyperparameter on Citeseer and Coauthor\_CS.
>
> * On Citeseer, the accuracy of **seen classes** shows a typical “rise-then-fall” trend as scale increases: small scale yields insufficient feature contrast; moderate scale enhances inter-class separability; overly large scale causes overfitting to seen classes. For **novel classes**, accuracy decreases overall with increasing scale, consistent with OpenIMA’s findings that stronger supervision biases the model toward seen classes. The slight rebound near scale = 10 likely comes from clearer seen-class decision boundaries, which indirectly benefit novel-class separability.
>
> * On Coauthor\_CS, **seen classes** follow the same “rise-then-fall” pattern. The **novel classes**, however, show a “fall-then-rise” trend. We attribute this to the following: small scale leads to weak supervised learning but still allows contrast-learning coarse boundary learning, giving relatively high novel accuracy; increasing scale strengthens the bias toward seen classes, reducing novel performance; when scale becomes very large, seen classes overfit, while novel classes—guided only by LLM pseudo-labels—benefit from stronger feature stretching and thus recover in accuracy.
>
> We will include this detailed analysis and figures in the revised manuscript (Figure 3b). Thanks again for your valuable suggestion!

---

> ### Author Response · Authors · 2025-11-21
>
> ## W3: Addition of a Notation Table
>
> **A3**: Thank you for your important suggestion. The important notations and their definitions are listed below:
>
> > Table 1: Summary of important notations with their definitions in this paper.
>
> | **Symbol**                                                   | **Definition**                                               |
> | ------------------------------------------------------------ | ------------------------------------------------------------ |
> | $\mathcal{G}=(\mathcal{V},\mathcal{E},\mathbf{X},\mathbf{A})$ | Graph with node set $\mathcal{V}$, edge set $\mathcal{E}$, feature matrix $\mathbf{X}$, and adjacency matrix $\mathbf{A}$. |
> | $N=\lvert \mathcal{V} \rvert$                                | Total number of nodes                                        |
> | $\mathcal{V}_l, \mathcal{V}_u$                               | Set of labeled nodes / Set of unlabeled nodes                |
> | $\mathcal{C}_l, \mathcal{C}_u$                               | Set of labeled (seen) classes / Set of unlabeled classes     |
> | $\mathcal{C}_n$                                              | Set of novel classes, $\mathcal{C}_n = \mathcal{C}_u \setminus \mathcal{C}_l$ |
> | $\mathcal{Y}_l, \mathcal{Y}_u$                               | Set of labels for nodes in $\mathcal{V}_l$, $\mathcal{V}_u$  |
> | $\mathcal{N}(v_i)$                                           | Neighbor set of node $v_i$                                   |
> | $U_*$                                                        | Node representations encoded by the GNN; $*\in\{1,2\}$ denotes different views. |
> | $\boldsymbol{w}_i, \boldsymbol{w}'_i$                                | Embeddings of node $i$ from the two views, used for contrastive learning. |
> | $\tau$                                                       | Temperature parameter                                        |
> | $\boldsymbol{m}_i, \boldsymbol{n}_i$                                 | Feature vectors from augmented embeddings                    |
> | $u(v_i)$                                                     | Uncertainty of node $v_i$                                    |
> | $r_{v_i}$                                                    | Influence quality score of node $v_i$                        |
> | $Q(v_j, v_i, k)$                                             | Influence of node $v_i$ on node $v_j$ after $k$ propagation steps |
> | $\text{score}(v_i)$                                          | Node selection score                                         |
> | $\mathcal{L}$                                                | A loss function                                              |
> | $y_i^{\text{LLM}}$                                           | Pseudo-label generated by the LLM for node $v_i$             |
> | $S$                                                          | Set of selected nodes for LLM annotation                     |
> | $y_j^{\text{prop}}$                                          | Propagated pseudo-label for node $v_j$                       |
> | $\hat{\mathcal{Y}}^t$                                        | Aggregated pseudo-label matrix at iteration $t$              |
> | $\hat{\mathcal{Y}}_{\text{new}}^t$                           | Newly generated pseudo-label matrix at iteration $t$         |
> | $\gamma$                                                     | Label decay coefficient                                      |
> | $\eta$                                                       | Scaling factor for cross-entropy loss $\mathcal{L}_{\text{CE}}$ |
>
>
> Thanks again for your constructive suggestion. We will include the above important symbol table in the appendix of the revised version to improve the paper's readability.
> ***
> ## W4: Addressing Typos and Notation Issues
>
> **A4**: Thank you for your careful review and corrections. We will amend "tau" to "$\tau$" in the revised manuscript and correct the other typographical errors in the revised manuscript.
> ***
> ## Q1: Detailed Explanation of the Soft Feedback Propagation
>
> **A5**: We thank the reviewer for the attention to the **soft feedback propagation (SFP) strategy**. The SFP mechanism is designed to suppress the propagation of noise introduced by the LLM via a soft-label approach. Specifically, the LLM first provides initial annotations for a subset of nodes. These labeled nodes then propagate their labels through the graph structure to generate **soft label distributions** for neighboring nodes. During this process, if the propagated soft label distribution of a labeled node deviates significantly from its original LLM label, the annotation is either **rejected or down-weighted**. In this way, SFP effectively prevents the accumulation and diffusion of LLM errors across the graph while maintaining high consistency in confident labels. Thanks again for pointing this out. We will clarify this in the revision.

---

> ### Author Response · Authors · 2025-11-21
>
> ## Q2: Clarification on the Calculation of the Influence Quality $r_{v_i}$
>
> **A6**: We thank the reviewer for the question. The influence quality $r_{v_i}$ in equation 8 is **not a predefined constant**; it is dynamically updated in each propagation round based on the current node features and oracle accuracy. As training progresses, node representations and semantic similarities evolve, enabling adaptive estimation of node influence quality.&#x20;
>
> Specifically, we introduce a **dynamic feature similarity and confidence-adjusted mechanism** to achieve adaptive computation. We first construct the **local propagation range** based on the graph’s two-hop adjacency and compute the **semantic similarity** between nodes:
>
> &#x20;$q_{ij} = \text{cosine}(p_i^{(2)}, p_j^{(2)})$
>
> where $p_i^{(2)}$ denotes the two-hop aggregated feature of node $v_i$. Then, integrating **structural similarity** and **label confidence**, the influence quality between nodes is defined as:
>
> $r_{ij} = \frac{\alpha_i q_{ij}}
>  {\alpha_i q_{ij} + (1 - \alpha_i)\frac{1 - q_{ij}}{C_N - 1}}$
>
> where $C_N$ is the number of classes and $\alpha_i$ reflects the confidence weight of the LLM annotation. The overall **influence quality** of a node $v_i$ is given by the average reliable influence over its two-hop neighborhood:
>
> $r\_{v\_i} = \frac{1}{|\mathcal{N}\_2(v\_i)|} \sum\_{v\_j \in \mathcal{N}\_2(v\_i)} \mathbb{1}\big[(A^2\_{ij} \cdot r\_{ij}) > \theta \big]$
>
> where $A^2\_{ij}$ denotes the two-hop adjacency matrix, $\theta$ is the activation threshold, and $\mathbb{1}$ is an indicator function.
>
> We appreciate this feedback and have revised the manuscript (Appendix H) accordingly.
>
> ***
>
> Thanks again for appreciating our work and for your constructive suggestions. We will properly include all the rebuttal contents in the revised version, following your valuable suggestions.

---

### Official Review · Reviewer_31nL · 2025-11-02

**Soundness:** 3
**Presentation:** 3
**Contribution:** 2
**Rating:** 4
**Confidence:** 4

**Summary:**

The paper studies the problem of leveraging LLMs for open world node classification. It aligns graph node embeddings with an LLM space, queries the LLM only on a small set of high value nodes selected by influence and uncertainty, and then propagates the LLM’s soft labels across the graph with safeguards to limit noise. On five benchmark graphs, LANO improves overall, seen, and especially novel class accuracy over strong open world baselines, and ablations show that removing LLMs or the selection strategy hurts performance.

**Strengths:**

1. The studied problem of open-world node classificaiton is important; using LLMs as flexible annotators to cover those emerging classes is a natural and well motivated direction.
2. The proposed framele work is reasonable with each step supporting the next one.
3. The reported gains over recent open world graph learners indicate that LLM supervision actually improves recognition of novel classes.

**Weaknesses:**

1. All evaluations are on relatively small and standard academic benchmarks. These are useful for controlled studies, but they may not stress test scalability like larger graph benchmarks such as those in OGB. Without such results it is hard to judge how the method behaves on real world sizes.
2. While the paper compares to many open world or open intent graph baselines, it does not run against the newest lines of work that also use LLMs as graph annotators or weak oracles [1]. Since those methods are conceptually closest, omitting them makes it harder to tell how much of the improvement is due to the specific active selection and bias controlled propagation introduced here, and how much is simply the benefit of calling an LLM at all.
3. The paper does not make the cost of using the LLM explicit. A budget table with number of calls, average prompt length, and wall clock per dataset would make the method much easier to adopt and would clarify its practical limits.

[1] Label-free Node Classification on Graphs with Large Language Models (LLMs). ICLR 2024

**Questions:**

1. Can the method run on a larger graph (e.g., OGB) without changing the selection strategy?
2. How many LLM queries did each dataset actually require?
3. How accurate are the LLM labels by confidence bucket, and how often does the model return “undecidable”?

---

> ### Author Response · Authors · 2025-11-21
>
> We are truly thankful for your insightful and constructive review. Our detailed responses are presented below.
> ***
> ## W1 & Q1: Experiments on a larger graph (e.g., OGB)
>
> **A1**: We thank the reviewer for raising the concern regarding **scalability**. In fact, for larger graphs (e.g., *Coauthor Physics dataset*), we adopt a random-walk–based sampling strategy combined with mini-batch clustering, enabling efficient training and representation alignment. The computational mode is already demonstrated in our experiments, showing that our method remains effective and stable across datasets of different scales.
>
> To address the reviewer’s concern, we have run additional experiments on **ogbn-arxiv (169K nodes)&#x20;**&#x75;sing the same open-world class split. We conducted three replicate experiments and calculated their average. Preliminary results show:
>
> > Table 1: Evaluation on larger datasets(ogbn-arxiv) by test accuracy (%). The best results in each column are highlighted in bold.
>
> | Method  | all  | seen | novel |
> | ------- | ---- | ---- | ----- |
> | ORCA-ZM | 41.6 | 47.0 | 31.6  |
> | ORCA    | 41.6 | 44.7 | 34.6  |
> | OpenCon | 32.2 | 31.8 | 31.6  |
> | OpenIMA | 43.6 | **49.2** | 32.9  |
> | LANO    | **50.1** | 39.1 | **53.1**  |
>
> It can be concluded that while classic benchmarks were used for controlled and fair open-world comparisons, the architecture of **LANO** is fundamentally **scalable**, and our preliminary OGB results already validate this. We will include these results in the revised version’s Appendix. Thanks again for your valuable comment!&#x20;
> ***
> ## W2: Comparison with LLM-based Graph Methods
>
> **A2**: We thank the reviewer for the comment. Although LLM-GNN\[1] and LANO both involve LLMs, their settings differ fundamentally: (1) LANO target&#x73;**&#x20;open-world node classification**, requiring detection and handling of novel classes, whereas LLM-GNN assumes a **closed-world** setup; (2) LLM-GNN operates on **Text-Attributed Graphs&#x20;**(TAGs), inserting raw node text directly into prompts with limited use of structure, while LANO does not rely on textual attributes—using node feature vectors, GNN-based encoding, and explicit GNN–LLM representation **alignment**; (3) LLM-GNN follows a label-free **zero-shot&#x20;**&#x70;aradigm, whereas LANO performs **open-world semi-supervised learning** with a labeled–unlabeled split, making direct comparison inappropriate. LANO is therefore **not&#x20;**&#x73;imply “LLM annotator + GNN,” but a full framework integrating alignment, influence- and uncertainty-aware annotation, and bias-reduced pseudo-label learning. Thus, direct comparison between LLM-GNN\[1] and our LANO across such distinct learning paradigms would therefore be unfair and potentially misleading.
>
> To address the reviewer’s concern, we compare experiments with the most relevant LLMs-as-graph-annotators method, LLM-GNN. Since LLM-GNN is originally designed under a **closed-world zero-shot** setting, we adapted it to the **open-world semi-supervised** setting to ensure a fair comparison. Specifically, we preset 50 labeled nodes, exposing only half of the classes as **seen** and treating the remaining classes as **unseen**. We then followed the LLM-GNN training pipeline—difficulty-aware active node selection, confidence-aware annotation, optional post-filtering, and GNN training & prediction.
>
> The final performance on the Citeseer dataset under this adapted setting is reported in Table 2.
>
> > Table 2: Comparison of LANO with other relevant work that also use LLMs as graph annotators, such as LLM-GNN, on CiteSeer dataset. The best results in each column are highlighted in bold.
>
> | Method  | all      | seen     | novel    |
> | ------- | -------- | -------- | -------- |
> | OpenIMA | 68.1     | 71.8     | 64.3     |
> | LLM-GNN | 66.9     | 66.3     | **67.5** |
> | LANO    | **70.2** | **73.8** | 66.2     |
>
> Based on the above experimental results and our ablation experiments, we confirm: (1) Most of LANO’s performance gain does not simply come from using an LLM. (2) Our active selection + bias-controlled propagation are the key contributors.
>
> Thanks again for this constructive comment.

---

> ### Author Response · Authors · 2025-11-21
>
> ## W3 & Q2: Addition of Cost Analysis of LLMs
>
> **A3**: We thank the reviewer for the constructive suggestion. We adopt a locally deployed Qwen3-8B model running on a single RTX 4090 GPU. For each dataset, we run the full pipeline three times consecutively and report the approximate statistics (average prompt length and wall clock) below:
>
> > Table 3: The LLM usage cost of LANO across datasets, including average prompt length (tokens) and wall clock (HH:MM:SS).
>
> | **Dataset**           | **Avg Prompt Length&#x20;**(tokens) | **Avg Wall Clock Time** |
> | --------------------- | ----------------------------------- | ----------------------- |
> | **Coauthor\_CS**      | 383                                 | 00:16:33                  |
> | **Coauthor\_phy**     | 372                                 | 03:14:07                 |
> | **Citeseer**          | 328                                 | 00:11:01                  |
> | **Amazon\_photos**    | 339                                 | 00:10:41                  |
> | **Amazon\_computers** | 350                                 | 00:21:01                  |
> | **ogbn-arxiv**        | 570                                 | 04:48:44                |
>
> **Number of calls**: In the early stages of training, the model has not yet learned strong encoding abilities, and the token representations still contain substantial noise. As a result, the LLM often outputs “uncertain” when queried, leading to unnecessary computational overhead. Therefore, we introduce LLM-based labeling only in the later stages of training.In each epoch, we query the LLM with 10 selected nodes. A typical training process contains around 40 epochs, so an entire experiment involves **approximately 200 queries**. However, we emphasize that this fixed number of epochs is used solely to verify that the training is sufficiently converged. If early stopping is enabled, the total number of queries could be significantly smaller.
>
> These results indicate that our approach remains computationally feasible even with a locally running 8B model. We will include the cost table in the revised version to make the practical computational budget fully transparent. Thanks again for your constructive comment.
> ***
> ## Q3: Confidence-based Accuracy Analysis and "Undecidable" Rate
>
> **A4**: We thank the reviewer for the question. We apply confidence-based thresholding to filter the LLM’s parsed outputs, using three thresholds: 50%, 75%, and 99%. For each threshold, we compute the LLM’s labeling accuracy using only the predictions whose confidence exceeds the corresponding cutoff. As shown in the table, the labeling accuracy consistently increases as the confidence threshold becomes higher, indicating that our threshold buckets effectively encourage the LLM to produce more reliable and robust predictions.
>
> > **Table 4.** *Annotation accuracy (%) of the LLM across different datasets at different thresholds (50%, 75%, and 99%).*
>
> | **Dataset**       | 50%    | 75%    | 99%    |
> | ----------------- | ------ | ------ | ------ |
> | citeseer          | 46.79 | 48.98 | 51.88 |
> | coauthor\_cs      | 31.58 | 33.33 | 35.71 |
> | amazon\_photos    | 27.91 | 33.14 | 38.04 |
> | amazon\_computers | 23.92 | 32.91 | 36.28 |
>
> **"Undecidable" Rate**: In our experiments, we did not explicitly log the number of “unable to determine’’ responses. However, based on available logs and backward estimation from the total annotations, we find that—with appropriate parameter configurations—the model returns an “unable to determine’’ label in approximately **25%–40%** of cases, depending on the dataset. This behavior is consistent with our design: the LLM refrains from assigning a label when its confidence score falls below the threshold, thereby maintaining the reliability of the generated annotations.
>
> Thanks again for your constructive suggestion. We will include the above results in the appendix of the revised version.
>
> ***
>
> In light of these responses, we hope we have addressed your concerns, and we hope you will consider raising your score. We will properly include all the rebuttal contents in the revised version, following your valuable suggestions.
>
> ***
>
> \[1] Label-free Node Classification on Graphs with Large Language Models (LLMs). ICLR 2024

---

### Author Response · Authors · 2025-12-04
**Summary for AC**

**Dear Area Chair,**

We truly appreciate the time and effort you are dedicating to reviewing our work under these exceptional circumstances.
To assist with your assessment, we provide a concise summary of our rebuttal and the status of reviewer concerns.

**Overall Status:**

Reviewers primarily raised concerns regardin&#x67;**&#x20;Comparison with LLM-based Graph Methods** and **Parameter sensitivity analysis**. In our rebuttal, we have provided comprehensive responses and added new experiments to address these points. We believe the major hurdles for acceptance have been cleared.

**Key Rebuttal Highlights:**

* **Concern 1: Comparison with LLM-based Graph Methods** (Raised by R1, R3)

  * **Action:** We have added comparisons with LLM-GNN\[1] in the revised version (Table 8) and have clarified the difference between our LANO and similar LLMs-as-graph-annotators work.

  * **Result:** Our method outperforms the baseline, demonstrating the effectiveness of our approach.

* **Concern 2: Parameter sensitivity analysis of scaling factor $\eta$** (Raised by R2)

  * **Action:** We included a sensitivity analysis plot of scaling factor $\eta$ in the revised version (Figure 3b).

  * **Result:** We analyzed the impact of different $\eta$ on the performance of seen and novel classes.

* **Concern 3: Typos and Notation Issues&#x20;**(Raised by R2, R3)

  * **Action:** We corrected all typographical errors in the revised manuscript and added a notation table in the Appendix C.

  * **Result:**  This significantly improves the paper's readability and help readers better understand the equations.

In the revised version, we have incorporated all reviewer responses and highlighted the revised sections in yellow for clarity. We hope this summary helps you navigate the discussion efficiently. We remain available for any further questions you may have. We sincerely thank you again for your careful assessment — your recognition would be tremendous encouragement for us!

Sincerely,

The Authors

---
[1] Zhikai Chen, Haitao Mao, Hongzhi Wen, Haoyu Han, Wei Jin, Haiyang Zhang, Hui Liu, and Jiliang Tang. Label-free node classification on graphs with large language models (llms). arXiv preprint arXiv:2310.04668, 2023.

---

### Meta-Review · Area_Chair_L2N8 · 2026-01-07

**Summary:**

This paper proposes LANO to leverage LLMs to solve the open-world node classification problem. LANO first aligns GNN representations with LLM token embeddings through instance-aware and feature-aware self-supervised learning, and then selects informative nodes for LLM to annotate cost-effective pseudo-labels.

The reviewers raised concerns about limited experimental scope, missing LLM-based baselines, presentation quality and so on. In the rebuttal, the authors addressed these issues by adding large-scale experiments on ogbn-arxiv, comparing with an adapted LLM-GNN baseline, reporting LLM cost and runtime, clarifying the prompt design, providing parameter sensitivity analysis and a notation table, and improving overall presentation.

However, some concerns remain. While the paper convincingly demonstrates the importance of open-world node classification, the experimental evaluation is still relatively limited in scale, and additional validation on larger, more diverse real-world datasets (e.g., anomaly detection scenarios) would strengthen the practical impact. Moreover, the technical novelty is somewhat incremental, as the use of LLMs for label annotation assistance has been explored in prior work.

Recommendation: Reject

**Reviewer Concerns:**

Reviewer 31nL raised concerns including the evaluation being limited to relatively small datasets, the absence of comparisons with recent LLM-based graph annotation or weak-oracle methods, and the lack of explicit reporting on API usage and time costs.

Reviewer QJvH questioned about the prompt design, suggested adding a table summarizing important notations and their definitions, and asked for explanations of the soft feedback propagation strategy as well as the computation of influence quality in Equation (8).

Reviewer cF53 expressed concerns about the limited novelty of the proposed approach, missing LLM-based baselines, insufficient experimental validation, and numerous presentation issues and typographical errors.

In response, the authors conducted additional experiments on the large-scale ogbn-arxiv dataset, added a variant of LLM-GNN as the baseline, reported API usage and time costs, clarified the prompt design, added a parameter sensitivity analysis for the scaling factor, included the notation table, explained the soft feedback propagation strategy and calculation of influence quality again, and substantially improved the presentation.

Some concerns about the breadth of large-scale evaluation and the degree of novelty relative to existing LLM-assisted graph learning methods remain partially open.

**Reviewer Scores:**

The rebuttal is unlikely to lead to significant score changes across reviewers.

---

### Decision · Program_Chairs · 2026-01-26

Reject